# Reconfigurable optoelectronic transistors for multimodal recognition

Pengzhan Li[1,2,7], Mingzhen Zhang[1,3,7], Qingli Zhou[2,7], Qinghua Zhang[1,4], Donggang Xie[1,3], Ge Li[1,3], Zhuohui Liu[1,5], Zheng Wang[1,3], Erjia Guo[1,3], Meng He[1], Can Wang[1,3], Lin Gu[6], Guozhen Yang[1], Kuijuan Jin[1,3] ✉ & Chen Ge[1,3] ✉

Biological nervous system outperforms in both dynamic and static information perception due to their capability to integrate the sensing, memory and processing functions. Reconfigurable neuromorphic transistors, which can be used to emulate different types of biological analogues in a single device, are important for creating compact and efficient neuromorphic computing networks, but their design remains challenging due to the need for opposing physical mechanisms to achieve different functions. Here we report a neuromorphic electrolyte-gated transistor that can be reconfigured to perform physical reservoir and synaptic functions. The device exhibits dynamics with tunable time-scales under optical and electrical stimuli. The nonlinear volatile property is suitable for reservoir computing, which can be used for multimodal pre-processing. The nonvolatility and programmability of the device through ion insertion/extraction achieved via electrolyte gating, which are required to realize synaptic functions, are verified. The device's superior performance in mimicking human perception of dynamic and static multisensory information based on the reconfigurable neuromorphic functions is also demonstrated. The present study provides an exciting paradigm for the realization of multimodal reconfigurable devices and opens an avenue for mimicking biological multisensory fusion.

Humans are bestowed with multi-sensory perceptions and understanding of complex and ever-changing environment[1–3], and the information obtained through vision and hearing accounts for more than 90% of the total information processed[4,5]. The external dynamic information with dimensional features is perceived and pre-processed by the eyes and ears[6,7], and then sent to the visual and auditory cortices for post-processing (Fig. 1a). The brain makes decisions and accumulates relevant experience based on the complementary information of the two channels. In this process of continuous experience and learning, the nervous system plays a vital role, with synaptic plasticity being an important foundation of understanding and adaptation. In contrast, the traditional computer architecture faces the bottleneck of high latency and large energy consumption induced by data shuffling between memory and processing units[8]. Therefore, it is difficult to cope with complex real-world tasks such as machine vision, autonomous driving, and human-machine interaction. Biologically-inspired artificial electronic systems are expected to solve this bottleneck, and have received considerable attention[9–12]. A promising approach is to

[1]Beijing National Laboratory for Condensed Matter Physics, Institute of Physics, Chinese Academy of Sciences, Beijing, China. [2]Key Laboratory of Terahertz Optoelectronics, Ministry of Education, Department of Physics, Capital Normal University, Beijing, China. [3]School of Physical Sciences, University of Chinese Academy of Science, Beijing, China. [4]Yangtze River Delta Physics Research Center Co. Ltd., Liyang, China. [5]College of Materials Science and Opto-Electronic Technology, University of Chinese Academy of Sciences, Beijing, China. [6]Beijing National Center for Electron Microscopy and Laboratory of Advanced Materials, Department of Materials Science and Engineering, Tsinghua University, Beijing, China. [7]These authors contributed equally: Pengzhan Li, Mingzhen Zhang, Qingli Zhou. ✉e-mail: kjjin@iphy.ac.cn; gechen@iphy.ac.cn

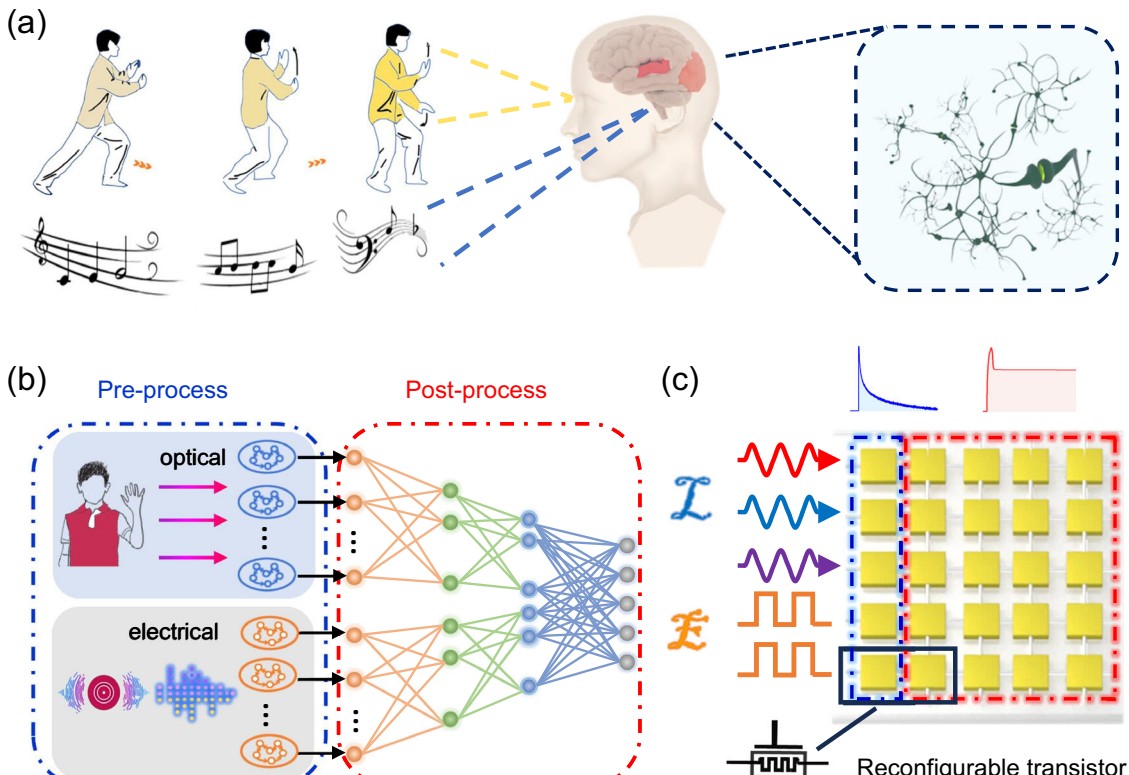

**Fig. 1 | Bioinspired neuromorphic audio-visual fusion system. a** Dynamic visual and auditory information obtained via the eyes and ears is transmitted to the relevant areas of the cerebral cortex. Biological neural network integrates two sensory information types to achieve high-level cognition. **b** A neural network architecture diagram for biologically inspired multi-sensory fusion. Left: Reservoir for preprocessing dynamic information. Right: Artificial neural networks (ANN) for high-level cognition and multimodal information integration. **c** Structural diagram of the hardware implementation of a multimode transistor with reconfigurable properties in response to light and electrical stimuli.

build a compact parallel optoelectronic fusion hardware system and simulate the audio-visual fusion process in the human brain[13–15].

Multimodal optoelectronic system can be divided into perception/pre-processing and post-processing core, which simulate the functions performed by human receptors and the cerebral cortex, respectively (Fig. 1b). The reservoir is generally used to pre-process input information with sequential characteristics[16,17]. A reservoir relies on nonlinear dynamics to convert low-dimensional signals into a high-dimensional state space[18–20], thereby enhancing computing efficiency and reducing time and energy consumption[21,22]. The hardware implementation of reservoirs should exhibit volatility, ensuring that the current state of a device is influenced by its recent experience without being affected by distant past events. Subsequently, pre-processed signals are conveyed to an artificial neural network (ANN) for information integration and inference. Artificial synaptic devices can replicate the plasticity of biological synapses, and in the hardware implementation of ANNs, they are responsible for post-processing and storage, necessitating non-volatility[23–25]. Due to the diametrically opposed dynamics required to realize volatile and non-volatile properties[26], it is difficult to incorporate the two behaviors in a single device. The implementation of the multimodal optoelectronic recognition system shown in Fig. 1b usually requires multiple units with separate reservoir and synaptic functions[27–29], which greatly increase the complexity of the circuit and integration. In previous studies, researchers generally used ideal software simulation models to replace artificial synaptic devices to perform weight storage and update functions in artificial neural networks[30]; or used sensors to convert different types of information into a single type of electrical signal for follow-up processing[29,31]. And information processing requires additional edge computing devices in most researches. This

essentially limits the potential for future applications of neuromorphic hardware systems. At present, realizing reconfigurable devices with integrated multimode sensing, memory and processing units (Fig. 1c), although desirable, remains a very challenging task[30–32].

Electrolyte-gated transistors (EGT), with tunable ion dynamic timescales at different gate voltages, show great potential for processing dynamic information in a single device[33,34]. Our previous study demonstrated that the reconfigurable property derives from the electric double layer (EDL) and the ion migration mechanisms[35]. In order to realize multimodal sensing, with this device structure design, $BaSnO_3$ (BSO) with tunable optical and electrical response was chosen as the channel material. Perovskite-structured BSO has attracted extensive attention due to its wide optical bandgap (~3.1 eV)[36] and high electron mobility at room temperature[37,38]. In addition to its reconfigurable characteristics under electrical regulation, BSO also has temporal and nonlinear memory decay behaviors under ultraviolet (UV)[39] that enables reservoir computing (RC). Therefore, due to its intrinsic properties with tunable multi-timescale optoelectronic response, electrolyte-gated BSO (BSO-EGT) may be a promising building block for multimodal reconfigurable transistors.

In this work, a BSO-EGT that integrates multimodal sensing, memory, and processing functions is introduced. The device can emulate switchable short- and long-term plasticity behaviors under optical and electrical stimulation. Time-scale modulation under UV light exposure originates from the generation of oxygen vacancies, while EDL and ion migration endow reconfigurable properties under voltage stimuli. Thus, both the reservoir and the neural network can be constructed based on the BSO-EGT. This device with multimode sensing and processing capabilities is used to recognize the Fashion-MNIST dataset containing multiple information. Due to its multimodal

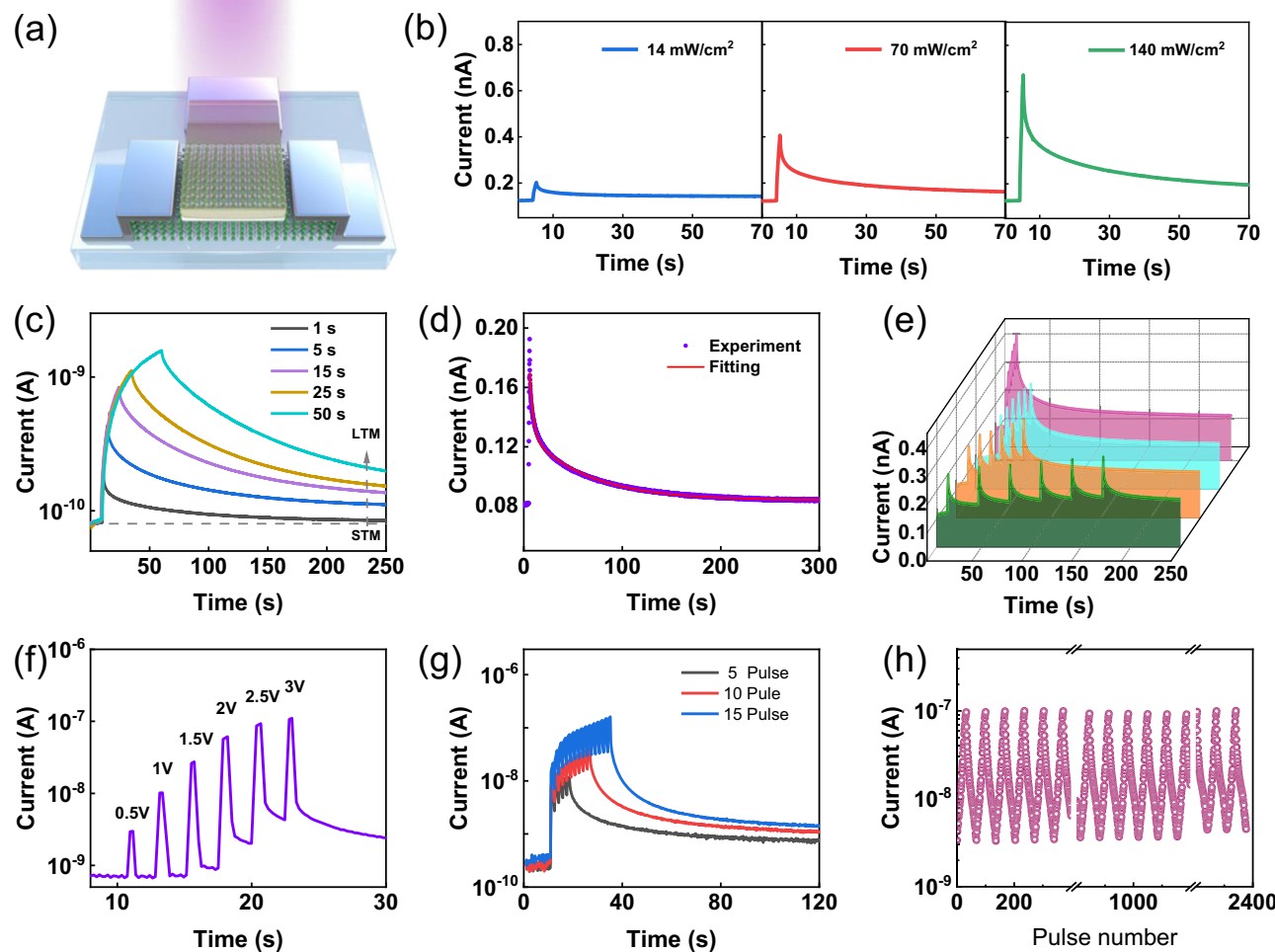

**Fig. 2 | EGT with reconfigurable characteristics for multimode sensing.**
**a** Schematic illustration of the neuromorphic transistor that can be stimulated using optical and electrical signals. The BaSnO₃ film serves as a channel between the source (S) and drain (D) electrodes, and IL is used as the gating medium. **b** Short-term potential effect stimulated using optical pulses (1 s) with a 0.3 V bias to monitor the channel current. **c** Transition from short-term memory (STM) to long-term memory (LTM) under different light intensities. **d** Memory decay process after light pulse (70 mW/cm² for 1 s) stimulation, fitted using a double-exponential function. The fitting result shows the coexistence of two physical processes.

**e** Excitatory post synaptic current (PSC) induced via a train of optical pulse (70 mW/cm², 1 s with intervals of 1, 5, 20, and 30 s) with a 0.3 V bias to monitor the channel current. **f** The channel current controlled through a series of $V_G$ pulse with a pulse width of 1 s and different amplitudes, +0.5 V, +1 V, +1.5 V and +2 V, which shows the transition of the PSC from volatile to non-volatile property. **g** Spike number-dependent plasticity under electrical stimuli (+2 V, 1 s) with the same intervals of 1 s. **h** Cyclic controlled LTP using $V_G$ (equally spaced from +1.5 to +3.5 V, duration of 1 s, spaced 1 s apart) and LTD using $V_G$ (equally spaced from −1.5 to −3.5 V, duration of 1 s, spaced 1 s apart) for 32 pulses.

---

nature, the fused information recognition exhibits higher accuracy than that achieved through single-signal processing. We further simulate the function of human audio-visual integration, demonstrating the potential to mimic the superiority of biological multisensory recognition, with an accuracy exceeding 90%.

## Results

### Tunable temporal dynamics under optical stimulation

We epitaxially grew high-quality BSO films on MgO (001) substrates using pulsed laser deposition (PLD). X-ray diffraction of the BSO showed a strong (002) peak accompanied by a (002) peak of MgO due to the epitaxial growth (Supplementary Fig. 1a), and the film thickness was 10 nm determined from the X-ray reflection (Supplementary Fig. 1b). Due to the large lattice mismatch ($\delta \approx +1.8\%$) between the film and substrate, reciprocal space mapping (RSM) around the MgO (204) peak showed that the BSO (lattice parameter, $a = 0.415$ nm) was relaxed on the MgO ($a = 0.421$ nm) substrate (Supplementary Fig. 2). High angle annular dark field scanning transmission electron microscopy (HAADF-STEM) and energy dispersive spectroscopy (EDS) results are shown in Supplementary Fig. 3. The atomic-level-resolved

STEM showed a clear interface between the film and substrate, and the BSO film mainly had a perovskite structure[40–42]. The corresponding EDS elemental mapping (Supplementary Fig. 3c–f) indicated the presence and spatial distribution of the BSO film and MgO substrate, and demonstrated a sharp interface. Then, the film was fabricated into an optoelectronic transistor, whose schematic diagram is shown in Fig. 2a. Supplementary Fig. 4 shows the optical microscopy image of the coplanar side-gate BSO-EGT and the left view of the device structure diagram. More details about the fabrication process of the transistor can be found in the "Methods" section.

The channel current $I_{SD}$ of the device was measured under different UV exposure conditions, and the scanning rate is 9 sampling points/second. The evolution of the channel current over time was monitored via the application of a reading voltage of +0.3 V. With a fixed pulse width of 1 s, the device exhibited volatile and decaying memory behavior when low-intensity light was applied (Fig. 2b). However, as the duration gradually increased, the optical response of the device changed from volatility to non-volatility, similar to the short-term memory (STM) to long-term memory (LTM) transition of biological synapses[43] (Fig. 2c). We also investigated the effect of

different light intensity on the $I_{SD}$ at the same light duration. As the intensity increased, the conductance exhibited the same transition and decay kinetics (Supplementary Fig. 5). To better describe the decay process of the current, a curve was used to fit the relaxation process after UV illumination, as shown in Fig. 2d. The current descent curve can be fitted well using a double exponential decay equation, as follow:

$$I(t) = C_1 \exp(-t/\tau_1) + C_2 \exp(-t/\tau_2) + C_0 \qquad (1)$$

where $C_0$, $C_1$, and $C_2$ are fitting coefficients, and $\tau_1$ and $\tau_2$ with fitted values of 1.635 s and 30.418 s are the characteristic time constants, which indicate the coexistence of rapid and slow relaxation in the descent process[44]. The rapid decrease in conductance after UV illumination can be attributed to the electron-hole pair recombination process in BSO[45], while the slow process is associated with the generation of oxygen vacancies and in-gap states. Oxygen vacancies ($V_O$) generated by UV irradiation are strongly localized on the film surface, which comes from the low mobility of $V_O$ in BSO films[39,46,47]. It can spontaneously return to the initial state after the stimulus is removed. As the light intensity increases, more oxygen vacancies are generated, leading to the appearance of non-volatility. Based on the above mechanisms, a paired pulse facilitation (PPF) effect can be achieved by applying a pair of UV pulse to the device (Supplementary Fig. 6).

The effect of pulse sequences consisting of eight pulses (pulse width 1 s, light intensity of 70 mW/cm²) at different intervals on the device behavior was also studied (Fig. 2e). The results show that shorter pulse intervals result in a higher current rise. The channel current can be stabilized and dynamically varied under the stimulation of a pulse sequence consisting of several pulses at different time intervals. The rise and decay of the channel currents reflect the temporal characteristics of the optical pulse stimulation, which is indicative of the device's high suitability for RC (Supplementary Fig. 7)[34]. Furthermore, the response under blue light (450 nm) illumination was investigated, and a similar volatile behavior as that under low-intensity UV illumination was observed, but with a much smaller magnitude of change (Supplementary Fig. 8). Optical transmittance spectra show strong absorption characteristics at wavelengths shorter than 400 nm (Supplementary Fig. 9a). The BSO films are transparent at visible and near-infrared wavelength due to large band gap of $E_g \approx 3.1$ eV (Supplementary Fig. 9b). Image recognition and memory are important functions of artificial vision systems. To simulate a UV-sensitive vision system, we fabricated a 3×4 pixelated array using BSO-EGT. Three letters "I", "O" and "P" are used to test the ability of the array to learn and remember images. Under the stimulation of 15 consecutive UV pulses, the image can still be clearly resolved after 350 s, which proves the advantages of our device in simulating the visual system. (Supplementary Fig. 10).

Previous studies have verified that oxygen vacancies will be generated in BSO films under UV irradiation, resulting in an increase of film conductance. Lee et al. used ambient-pressure X-ray photoemission spectroscopy (APXPS) for in situ characterization to monitor the origin of generated defects and proved that the UV illumination under vacuum leads to chemical modification by evolution of oxygen-vacancy-related defects on the surface of BaSnO₃[39]. And the result of cross-sectional scanning transmission electron microscopy (STEM) in vacuum-illuminated BaSnO₃ epitaxial films also confirmed the existence of oxygen-related defects at the surface[39]. When the UV dose [UV dose (mJ/cm²) = UV Intensity (mW/cm²) × Exposure Time (s)] is low, the oxygen vacancy concentration is low, and its impact on the BSO channel can be recovered quickly. On the other hand, when the UV dose is high, the oxygen vacancy concentration increases, resulting in non-volatile conductance changes in the BSO channel.

## Reconfigurable dynamics of BSO-EGT under electric stimuli

Next, we examined the electrical modulation of BSO through electrolyte gating. Due to the powerful regulation capability of ionic liquids (ILs)[48], the BSO-EGT was constructed using N,N-diethyl-N-(2-methoxyethyl)-N-methylammoniumbis-(trifluoromethyllsulphonyl)-imide (DEME-TFSI) as the electrolyte-gating medium. Supplementary Fig. 11 shows the transistor characteristic curves of BSO-EGT, namely the transfer characteristic curve and the output characteristic curve. Supplementary Fig. 11a illustrates the transfer curves measured along the counterclockwise direction, and the scan rate is 10 mV/s. When the gate bias is +2.5 V, the channel current is 0.75μA and the leakage current is 0.8 nA. The difference between the two reaches three orders of magnitude, so the effect of leakage current can be neglected. The response of the device to electrical pulse stimulation was then measured at $V_{SD} = 0.3$ V. At low $V_G$, the channel current can quickly decrease to initial state, but the current did not fully recover as the increase of $V_G$ (Fig. 2f). That is, the device transitioned from volatility to non-volatility with the increasement of $V_G$, exhibiting typical reconfigurable characteristics. Then, a voltage bias with fixed amplitude (+1 V) and variable pulse width was applied to the gate. The device exhibited volatility, with the current rapidly decaying to initial state when the low $V_G$ is removed (Supplementary Fig. 12a). During electrical stimulus, the scanning rate is 5 sampling points/second. The volatile response of BSO-EGT at a lower $V_G$ originated from the rapid movement of anions and cations in ILs under an electric field, which produced a strong accumulation of space charge at the interface between the electrolyte and the channel, called a Helmholtz layer or EDL[49]. Accumulation occurs due to blocking ions in the solid channel. In order to balance the EDL formed at the interface, an accumulation of electrons occurs inside the channel. Therefore, the conductance of the channel will change. When the external bias voltage is removed, the ions at the interface spontaneously migrate back into the ionic liquid, and the EDL disappears, so the channel conductance returns to its original state[35].

But, under the stimulation of voltage pulses $V_G = +2$ V with different durations, the conductance of the device can be maintained at a high level without returning to initial state (Supplementary Fig. 12b). Moreover, the correlation of device characteristics with gate voltage pulses of different time intervals, from 0.1 s to 2 s, was tested by applying a pulse sequence (+1 V, 1 s). As the interval decreased, the level of current accumulation became more pronounced (Supplementary Fig. 13). This is due to the fact that only a small part of the ions accumulated at the interface relaxes back into the bulk when the spacing is short, thereby promoting channel doping and conductance. The device conductance can be maintained at different levels as the number of applied pulses increases under $V_G$ of +2 V (Fig. 2g). To explore the multi-level memory properties of the device, $I_{SD}$ was adjusted to different conductance levels, and the retention characteristics were tested. There was no significant decay of channel conductance over a period of 300 s (Supplementary Fig. 14), indicating that BSO-EGT has good non-volatility. Furthermore, the pulse-switching characteristics of electrical potentiation (+2 V, 1 s) and depression (−2 V, 1 s) were investigated. The transistor was reversibly switched between the high- and low-conductance states hundreds of times without significant degradation (Supplementary Fig. 15).

The conductance of the BSO channel showed non-volatile changes under high voltage stimulation, and there is a peak in the gate current of the transfer characteristic curve at -1.3 V (Supplementary Fig. 11a), which is related to the hydrolysis reaction[50]. This is because, in addition to the presence of EDL at the interface, Protons (H⁺) originating from the trace water containing in the ionic liquids would be injected into the film when the positive voltage exceeded the critical voltage. Protons produced by hydrolysis can be driven to the channel interior, resulting in strong interactions with the solid material[49,51]. Therefore, the non-volatility of the device came from the migration of protons generated by hydrolysis in the ionic liquid into the oxide film

under the positive gating, causing a non-volatile increase in the channel conductance.

To verify this mechanism, secondary ion mass spectrometry (SIMS) was performed on the BSO films after applying different electrical stimuli. Supplementary Fig. 16a shows that hydrogen ions appear inside the electrically modulated films, and the hydrogen ion concentration rise significantly as the voltage increased. We added a small volume of $D_2O$ to the IL, in which $D^+$ just acts as an isotope marker to show the source of the $H^+$. The addition of $D_2O$ to the IL resulted in the presence of $D^+$ signal in the electrically-modulated film, which was not observed inside the film without modulation (Supplementary Fig. 16b). Therefore, the non-volatility of the BSO-EGT comes from the injection and diffusion process of hydrogen ions originating from hydrolysis. Additionally, there is a 12-hour waiting time for ionic liquid-gated samples before SIMS experiments were performed. Therefore, this also indirectly reflects that the insertion of hydrogen ions can exist inside the sample for a long time, causing a non-volatile effect. Long-term synaptic plasticity including long-term potentiation (LTP) and long-term depression (LTD) were simulated using our transistor (Fig. 2h). LTP was simulated using 32 consecutive positive electrical pulses (voltages from +1.5 to +3.5 V with a pulse width of 1 s), while LTD appeared when 32 negative voltage pulses were applied to the gate (−1.5 to −3.5 V, the pulse width is 1 s). The above result means that reconfiguration between volatile and non-volatile transistors can be achieved through electrical stimuli by controlling $V_G$.

## Multimodal characteristic of optoelectronic BSO-EGT

The temporal dynamics of the BSO-EGT under separate optical and electrical stimulation, provide two means of modulating the device characteristics. Furthermore, the fused optoelectronic response characteristics of the BSO-EGT devices are analyzed. There are three combinations of dual pulse stimulation: two electrical pulses (EE), one light and one electrical pulse (LE), and two light pulses (LL). The evolution of $I_{SD}$ under various conditions is shown in Supplementary Fig. 17a, where the width of both optical and electrical pulses is 1 s. The sampling point was the current value after a delay for 1 s from the second pulse. And the scanning rate is 9 sampling points/second. It can be seen that under the three different stimulation modes, the current of the device reaches three distinct states. This is because the application of a light pulse followed by an electrical pulse induces a further increase in conductance, and the two kinds of stimuli have different relaxation dynamics. Therefore, the response to multimodal inputs exhibits different decay characteristics compared to those obtained under individual light or electrical stimuli. Supplementary Fig. 17b shows the effect of the time interval between the optical and electrical pulses on the conductance state of the BSO-EGT. As the interval time increases, the collected current value decreases significantly.

In order to evaluate the multimodal sensing capability of optoelectronic reservoirs, 4-bit binary streams of different fused input modes were applied. Classified into five combinations, namely "LLLL", "EEEE", "LLLE", "LLEE", and "LEEE", respectively. As an example, mode "LLEE" denotes that the first two stimulation pulses are optical and the last two are electrical. As shown in Fig. 3a, each square wave input is considered as one bit, and the 4-bit input stream is encoded as a pulse sequence from "0000" to "1111", in which the "off" and "on" state of the optical or electrical pulse denote "0" and "1", respectively. Figure 3b shows the temporal variation of the drain currents for four different inputs, indicating that the final state of the reservoir is not only dependent on the last stimulus, but also related to the history of external stimuli. Figure 3c illustrates the evolution of the channel current after each pulse application for 16 combinations of inputs. Due to the nonlinear relaxation characteristics of the reservoir, its final state depends on its activity history. Therefore, "0001" and "0010" are two different sequences for the reservoir. When the input mode is "LLLE", the final states can be distinguishable from each other even

with the same initial states (Fig. 3d). The distinguishability of the 16 states indicates the potential of our device for in-memory RC applications[52]. When only one type of external stimulus was applied ("LLLL" and "EEEE"), the BSO-EGT also showed good distinguishability between inputs with different coding sequences (Supplementary Fig. 18).

The distributions of 16-state values for different modes are distinguishable, so the classification tasks can ultimately be performed through computer simulation. The classification results for the other two multimodal inputs ("LLEE" and "LEEE") were documented in Supplementary Fig. 19. It should be noted that as long as the relaxation process of the device has nonlinear and volatile characteristics, in theory, whether the potentiation or depression curves can be used to implement RC. Our as-prepared BSO-EGT is in a high-resistance state, the downward adjustment in the pristine state will cause significant leakage, making it difficult to perform RC based on the depression process. But there have been recent studies utilizing the depression behavior to implement reservoirs[53,54].

## Static image recognition with reconfigurable BSO-EGT

Human perceive external information through a multi-sensory approach, while existing electronic systems use a relatively single method with a limited range of applications. A multi-sensory fusion method can be adopted to extend the application scope. Since the BSO transistors can respond to both optical and electrical inputs, it is possible to read contaminated (i.e., noisy or partially complete) image information using a hybrid optical-electrical approach. Furthermore, BSO-EGTs demonstrate both volatility and non-volatility in response to various external stimuli. The volatile nature, coupled with the non-linearity of the device, offers great potential as a reservoir for image pre-processing. In contrast, the non-volatile property fulfills the requirements of artificial synapses, as it facilitates post-processing and storage functionalities[16,30]. Based on the above-mentioned reconfigurable and optoelectronic sensing properties of BSO-EGT, we extracted the specific parameters of the device and further simulated a system integrating reservoir and ANN functions (Supplementary Fig. 20). Using this system, we first investigated the recognition of static polluted images.

Fashion-MNIST dataset were chosen for demonstration (see Supplementary Fig. 21 for image samples). For pictures of Fashion-MNIST partially contaminated by pigments (Fig. 4a), the left side can be perceived through optical signals, whereas the right side necessitates the assistance of pressure sensors to convert tactile sensations into electrical signals. Here, the contaminated part is defined as invisible information. The "pollution degree" or "invisible degree" indicates how many proportions of the pixels in the picture are contaminated. For ease of demonstration, we omit the sensing and processing steps of the piezoelectric sensors. The image information of the polluted part can no longer be obtained through optical perception, but can only be obtained through electrical signals. The detailed processing flow for contaminated images is shown in Fig. 4a. The original image ($28 \times 28 = 784$ pixels) is reorganized into a stream of $196 \times 4$ pixels, after which the average value between pixels is used as the threshold to binarize the images. The processed image is read with two single-signal modes ("LLLL" and "EEEE") and three mixed-signal modes ("LLLE", "LLEE" and "LEEE"), respectively. The 196 groups of inputs are fed into the corresponding reservoir for pre-processing, which can compress the amount of data and map the low-dimensional feature space into a high-dimensional one using the nonlinearity process of the devices to facilitate the subsequent classification[18,55].

An artificial neural network (ANN) is used for training and inference with the weight update model based on the experimental measurements. Figure 4b shows the outputs of the "LLLL" reservoirs with contamination degree of 10% (top panel) and 90% (bottom panel), respectively. Obviously, the network operating under the "LLLL" read

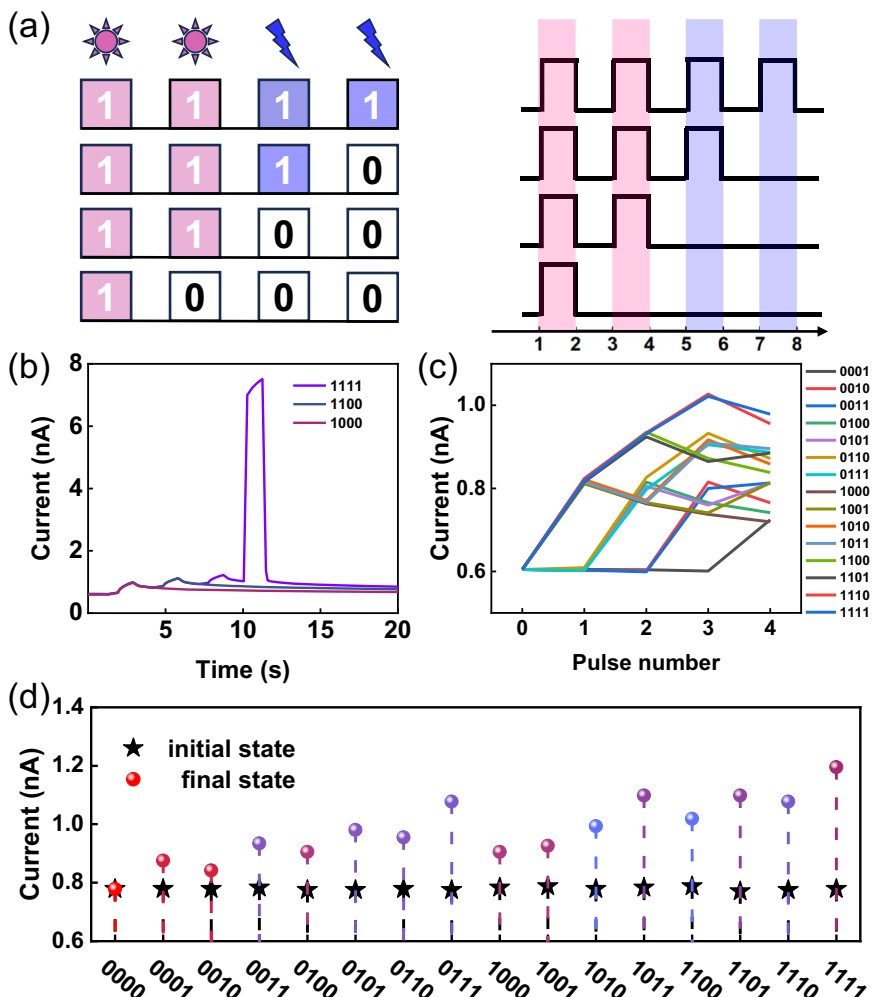

**Fig. 3 | Nonlinear mapping of multimodal signals based on EGT reservoirs.** **a** Combination mode for 16 inputs of 4-bit optical and electrical pulses. Here, both the optical and electrical pulse widths were 1 s and the pulse interval was 1 s. **b** Response characteristics of the channel current in 1111, 1110, 1100, 1000 combinations under the "LLLE" mode input (light intensity 70 mW/cm$^2$, $V_G = 1$ V). **c** The distinguishable output of 4-bit reservoir states reading at 0.3 V. **d** Initial and final current values of the reservoir state with pulse stimulation of 16 combinations in the "LLLE" inputs mode.

mode can obtain little effective information when there is a high percentage of contamination. Supplementary Fig. 22 gives the results of the other four read modes. Figure 4c illustrates the distribution of ANN synaptic weights before and after training. After the training process, the synaptic weights change from a random to a normal distribution, indicating that the neural network has been trained effectively. Figure 4d–f presents the final recognition accuracy obtained by the five read modes under different contamination levels. An invisibility of 0% means that the reservoirs of "LLLL" mode can acquire all the information from the image, while that of "EEEE" mode cannot obtain any content. Therefore, the "LLLL" mode is able to achieve a higher recognition rate as the classification is based on the actual information, while the inference of the "EEEE" model is indistinguishable from random guessing, so its recognition accuracy is much lower (Fig. 4d). And the situation is completely opposite in the case of 100% invisibility. The single-signal modes do not facilitate the effective image information acquisition in particularly extreme cases, while the mixed-signal modes allow for judgment and classification based on actual information due to its utilization of two reading channels. Thus, "LLLE", "LLEE", and "LEEE" can complete image recognition tasks with around 90% recognition accuracy at all contamination levels. When just one channel is used to obtain information, only parts of the features can be obtained, and the recognition accuracy is low. However,

the mixed mode can collect more comprehensive and rich information, so a higher recognition accuracy can be obtained. This result indicates that the mixed modes are more universal and can be applied to a wide range of complex situations.

### Multimodal dynamic gesture recognition with reconfigurable BSO-EGT

As a proof of concept, we mimic human perception of dynamic gesture recognition based on reconfigurable BSO-EGT. For gestures with spatiotemporal information, the decoupled sensation of vision and hearing may lead to misjudgment the direction or the object. If the two modes are combined for audio-visual fusion perception, the correct recognition accuracy can be greatly improved (Fig. 5a). Here, the EgoGesture dataset[56,57] was employed. Five gestures (Supplementary Fig. 23) were selected from eighty-three categories, and a sub-dataset with a sample size of 1250 was constructed. More details about dataset construction can be found in the "Methods". Here, each sample contains four frames. Figure 5b represents the three-dimensional spatial map of the first one, in which the XY plane refers to the coordinates, while the Z-axis is the value of pixels. The color of the data points is a linear mapping of the pixel values, which enables the content of the picture to be analyzed based on the distribution of the color and position of the orbs. Supplementary Fig. 24 gives the spatial maps of

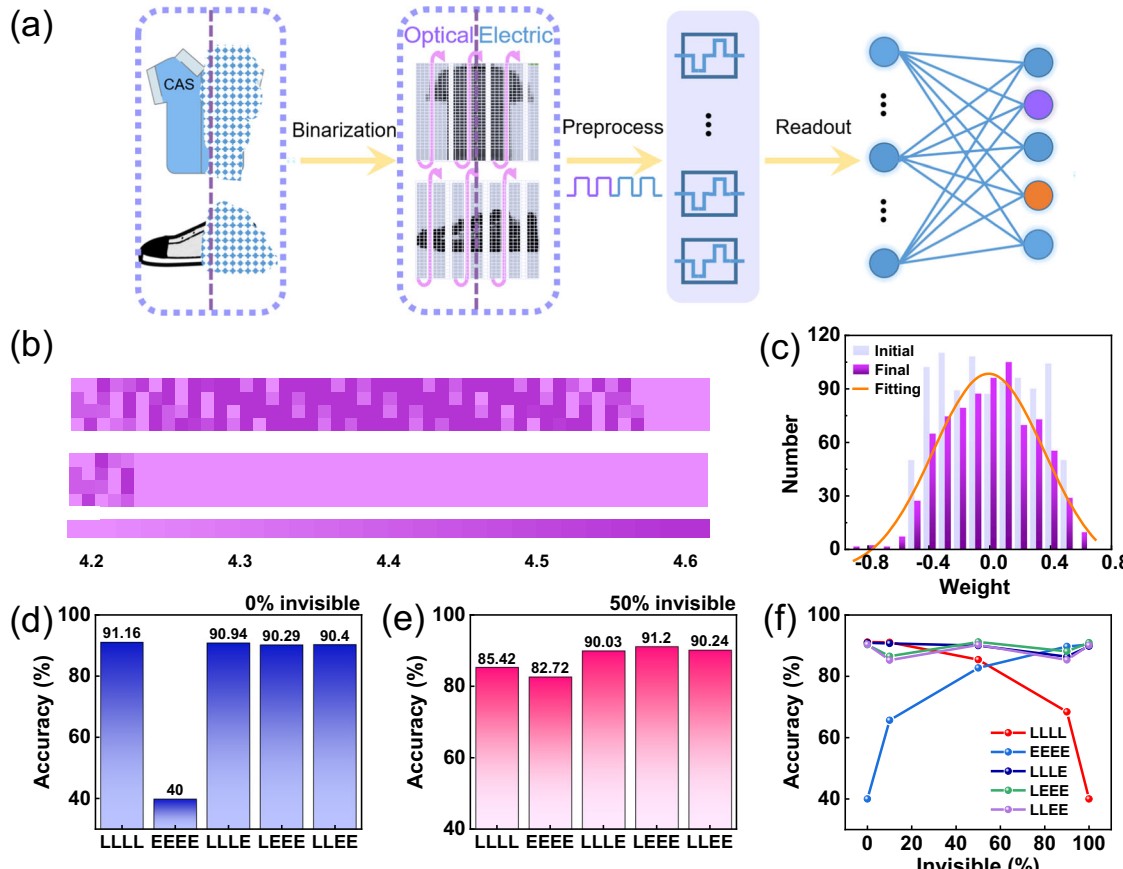

**Fig. 4 | Fused information input reservoir for recognition of multimodal Fashion-MNIST datasets. a** Image perception in complex environments. The right half of the information is invisible which can only be perceived through electrical signals, while the left half of information is visible which can only be perceived through visual signals. The original image ($28 \times 28 = 764$ pixels) was reorganized into a $196 \times 4$ by column vector array. **b** Output results of the reservoir in the "LLLL" sensing mode, when 10% (top) and 90% (bottom) of the image can only be read by electrical signals. **c** The change of ANN synaptic weight before and after training. The trained synaptic weight conforms to the normal distribution. **d, e** Recognition accuracy under the five perception methods when 0% and 50% of the image can only be collected by electrical signals. **f** Recognition accuracy of the five information perception modes variation with the proportion of the contaminated part (which can only be read via electrical signals).

the other three frames of this sample. The four figures have large similarity, indicating that the contents of the four frames are similar. However, the color and spatial distributions of the data points differ in details, suggesting that they are images of different states of moving objects.

Similar to the static image processing process, the multi-frame pictures with temporal information can be mapped to the pixel matrix into a light pulse matrix, which is illuminated on the optical reservoir array to achieve information perception. More details can be found in "Methods". The visualization diagram for the inter-mediate state shows the result of the superposition of the four frames after the pre-processing of the reservoirs (Fig. 5c). The last light pulse stimulation can be retained to the greatest extent, so the information of the last picture is clearer than that of the previous three. Furthermore, the overlapping part of the wrist in the four pictures will be stimulated with four consecutive pulses, so the output signal of the overlapping part was the strongest. Afterward, the pre-processed data would be fed into the neural network for subsequent training and inference. To avoid the slow training of processing units caused by simulation based on device parameters, the ANN in this part is built with the ideal model, while the reservoir is constructed based on device parameters.

Through 25 different public AI sound sources and text-to-speech conversion program, the names of the five selected actions are converted into speech, and 125 original samples are generated. Then, these samples were Fourier-transformed to obtain the frequency domain information. Figure 5d shows the speech spectrogram of a sample, which reflects the frequency domain distribution of the audio at different time moments. The time-domain and frequency-domain information were sampled separately and combined into an input vector of length 2000. On this basis, 10% random noise is added to the audio input signals to simulate possible bit error ratio (BER) during digital signal transmission, and the number of samples in audio dataset was also expanded to 1250 to match the size of corresponding video dataset. More details about dataset construction can be found in the Methods sub-section "Dataset construction". Afterward, the operations consistent with Fig. 4 were carried out, and Fig. 5e shows the input vector and the reservoir output.

After 3000 epochs, the accuracy of both video and audio recognition is lower than 80%, indicating that it is difficult to recognize different gestures with a single modal information (Fig. 5f). Since the BSO-EGT has the property of responding to multiple stimuli, a multi-sensory fusion neural network[29,58] using decision fusion method was constructed to demonstrate the multimodal gesture recognition. The audio and video datasets are processed through their respective reservoirs and neural networks, and then integrated and analyzed through the fusion-layer neural network. Evidently, the multimodal recognition method greatly improves the gesture classification

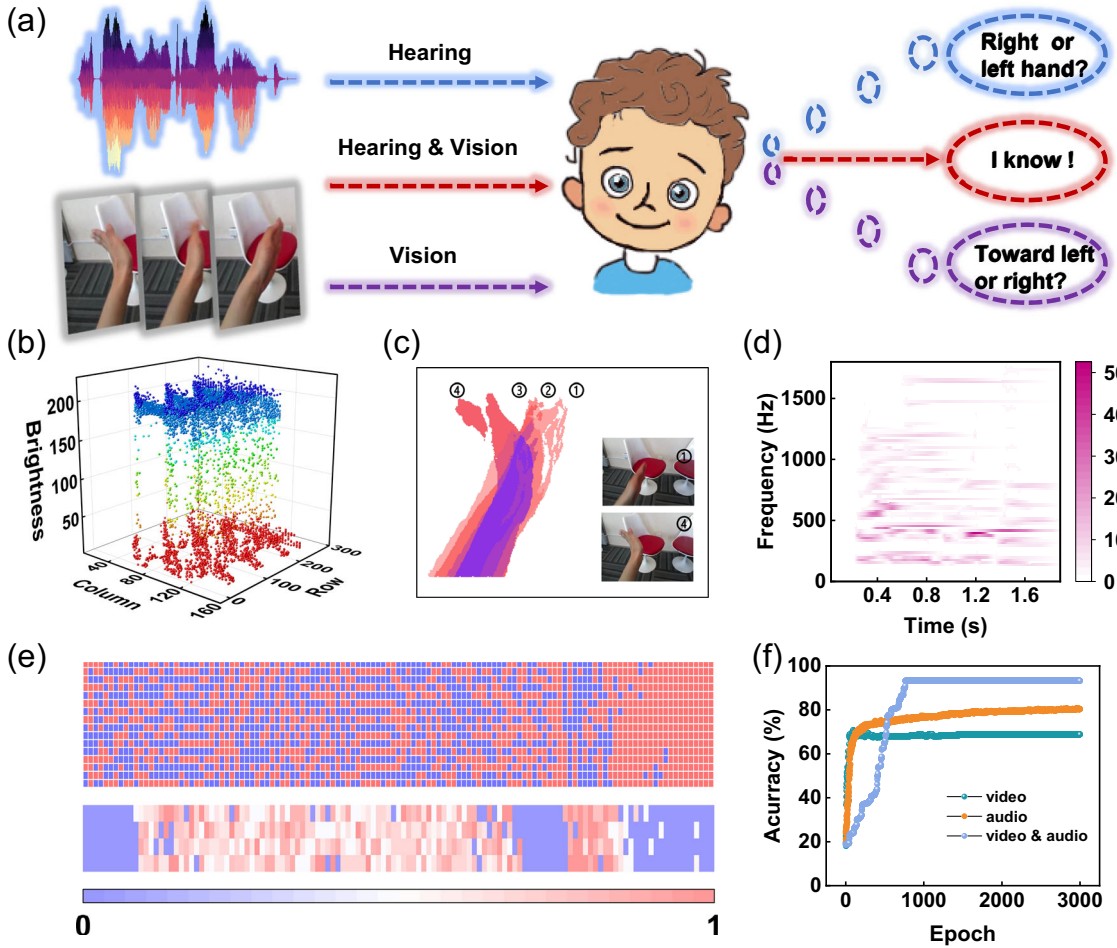

**Fig. 5 | Mimics human perception of dynamic audio-visual information.**
**a** Simplified diagram of humans achieving high levels of cognition through audio-visual integration. **b** Three-dimensional space mapping of "Rolling hand toward left" gesture, a sample from the EgoGesture dataset. **c** Images superimposed at four time points in the intermediate state after hand movements were preprocessed by the reservoir. **d** Speech spectrogram of "Rolling hand toward left" gesture, reflecting the frequency domain distribution of audio at different moments. Corresponding voice overs for gestures, generated using text-to-speech engine. **e** Input vector after normalization of time and frequency domain information, and output vector after reservoir processing. **f** Comparison of recognition accuracy under single- and multi-mode information processing.

accuracy, which reached 94.24% after 3000 epochs (Fig. 5f). This result indicates that the multimodal recognition method based on reconfigurable BSO-EGT has superior performance. Compared with previous studies, our BSO-EGT has the advantages of precise regulation and comprehensive functions in terms of reconfigurability (Supplementary Table 1). At the application demonstration level, we verified its multisensory integration capabilities and realized advanced neuromorphic applications with the multimode sensing, storage, and processing capabilities of a single device.

**Demonstration of wide application wavelength based on IGZO**
In this work, we mainly propose a design principle for a kind of reconfigurable device. Channel materials include but are not limited to BaSnO₃. A variety of oxide materials with responses to light can serve as channels. For example, InGaZnO₄ (IGZO), which is widely concerned by industry, can also be an option. IGZO has been mainly applied in industry due to its optical transparency, low processing temperature, and compatibility with various gate insulators[59,60]. It occurs the band-to-band excitation, the oxygen vacancies ionization, and the meta-stable peroxides formation in the a-IGZO semiconductor during the light illumination[61,62]. Due to the high density of the trap states in the gap, the high-quality a-IGZO films have a small conduction band tail (-2.3 eV)[63], resulting in broad spectral responses. The use of amorphous IGZO in thin film transistors offers numerous advantages, including excellent uniformity, high mobility, high switching current ratio, and large-scale processing[64].

We grew 15 nm a-IGZO film on SiO₂/Si substrates and X-ray photoelectron spectroscopy (XPS) results showed clear characteristic peaks of In 3*d*, Ga 2*p*, and Zn 2*p*, respectively, indicating that the IGZO film is of high quality (Supplementary Fig. 25). IGZO was used as the channel material to fabricated an electrolyte-gated transistor (IGZO-EGT) with the same structure. Base on the same experiment scheme, IGZO showed the potential to achieve the same functionalities we demonstrated with BSO film. IGZO-EGT also demonstrates the ability to accurately switch between volatile and non-volatile modes affected by voltage amplitude (Supplementary Fig. 26). The response of IGZO-EGT at green (532 nm), blue (450 nm), and ultraviolet (375 nm) light wavelengths was also tested (Supplementary Fig. 27) and showed the wide applicable light wavelengths of IGZO-EGT. Moreover, a clear nonlinear relaxation process was observed in IGZO-EGT after the electrical or optical stimulation was removed, indicating its potential to realize reservoirs. Therefore, using the design concept we proposed, electrolyte transistors using IGZO as the channel material can also achieve reconfigurable reservoir and artificial synapse functions, and then complete complex tasks such as audio-visual fusion recognition.

## Discussion

We have reported a neuromorphic EGT with multimode sensing, memory, and processing capabilities. The devices exhibit dynamical processes with tunable time scales under optical and electrical modulation, and are capable of realizing reconfigurable functions between physical reservoir and artificial synapse. A parallel optoelectronic fusion system composed of reservoir and ANN functions was simulated based on reconfigurable BSO-EGT. The Fashion-MNIST dataset containing multiple types of information was utilized as a standard test, and the recognition accuracy above 90% shows the superiority of this system in information processing. Furthermore, dynamic gesture recognition was also used to test the system performance. The higher recognition accuracy under audio-visual integration demonstrates the advantages of our constructed system in simulating biological multisensory fusion. Moreover, the approach demonstrated in our study can be utilized to a broad range of materials, as long as its relaxation process under stimuli has non-linear and volatile characteristics. Benefiting from the multi-modal sensing, storage and processing performance of our proposed device, complete audio-visual integration and recognition tasks can be realized on a single device. The processing of dynamic tasks reflects the real-time processing capability of our proposed device. The proposed system could advance the development of a multi-sensory human-machine interaction platform.

## Methods

### Sample preparation

The 10 nm $BaSnO_3$ film was epitaxially grown on (001)-oriented MgO substrates at 780 °C under $O_2$ pressure of 5.5 Pa. Pulsed laser deposition was used with a 308-nm XeCl excimer laser, with an energy density of about 1 J/cm² and a repetition of 3 Hz. The samples were cooled down to room temperature at 20 °C/min. The growth conditions were optimized to minimize the cation non-stoichiometry induced defects of as-grown $BaSnO_3$ films by adjusting the stoichiometric accuracy ([Sn]/[Ba] = 1) through a suitable target-to-substrate distance ($d_{ts}$ = 50 mm).

The 15 nm a-IGZO film was grown on $SiO_2$/Si substrates at room temperature under $O_2$ pressure of 5 Pa through pulsed laser deposition. Afterward, the film was annealed at 300 °C for 1 h. Pulsed laser deposition was equipped with a 308-nm XeCl excimer laser, with an energy density of about 1 J/cm² and a repetition of 3 Hz. The samples were cooled down to room temperature at 20 °C/min.

### Device fabrication

Through photolithography and ion beam etching technology, the BSO film is patterned into channels with an effective area of $50 \times 180$ μm². A coplanar side-gate structure is adopted, and a 30 nm Pt layer is deposited as an electrode by magnetron sputtering. The distance between the gate and the channel is 10 μm. The transistor device was completed by dropping an ionic liquid N, N-diethyl-n-(2-methox-yenthyl)-N-methylammoniumbis-(trifluoromethylsulphonyl)-imide (DEME-TFSI) on the channel and gate electrodes.

### Material characterization

X-ray diffraction patterns of the $BaSnO_3$ film was performed using a Rigaku SmartLab instrument with a 2θ range from 35 to 50° in step of 0.05°. STEM imaging was conducted by a double Cs-corrected JEOL JEM-ARM200CF operated at 200 kV with a CEOS Cs corrector (CEOS GmbH, Heidelberg, Germany). HAADF-STEM images were recorded with collection semi-angles of 90−370 mrad. Optical transmittance spectra were taken in air at room temperature with spectro-photometers (Cary 5000 UV-Vis-NIR, Agilent and Excalibur3100, Varain). In order to reveal the relationship between hydrogen concentration and gating voltages, a TOF-SIMS system (ION-TOF Gmbh)

was used to identify the depth profiles of protons. XPS measurements were performed on ThermoFisher Scientific ESCALAB 250X under monochromatic Al Kα radiation with an energy of 1486.6 eV.

### Device characterization

All the electrical characterizations were measured in a Laskeshore probe station with a Keithley 4200 semiconductor parameter analyzer in vacuum at room temperature. An UV laser at a wavelength of 375 nm and a blue laser at 450 nm were used for optical excitation in the experiments.

### Dataset construction

The video classification task uses the public EgoGesture dataset, which contains a total of 83 different gesture. Each category contains a large number of samples and each sample in this dataset contains dozens of chronologically arranged frames describing the corresponding gesture. We selected 5 gestures from 83 categories and the first 250 samples from each corresponding category are retained to construct a sub-dataset with a sample size of 1250.

The audio dataset is generated based on the built video sub-dataset. Through 25 different public AI sound sources, the names of the five selected gestures from EgoGesture dataset are converted into speeches, and 125 original samples are generated. After pre-processing and adding noise, the volume of the audio dataset also expands to 1250. In the video and audio datasets, the ratio of training set and test set is 1:1 which means each category contains 625 samples.

### Video classification

Each sample in the EgoGesture dataset contains dozens of frames. To compress the amount of data, we randomly extracted four frames in chronological order. The size of each frame is 320 pixels × 240 pixels, which can be represented by a binary matrix of size [320, 240] after binarization. Then, we expanded the matrix to get a one-dimensional column vector with the size of [76,800, 1]. After merging the vectors corresponding to the four frames, every final input needs to be represented by a matrix of size [76,800, 4]. Then, each row of the matrix was encoded with light pulses and fed into the corresponding reservoir. After being processed and compressed by the reservoirs, the input of the ANN is a vector of length 76,800. Therefore, a two-layer fully connected neural network with a size of [76,800, 5000, 5] was used to perform the video classification recognition task. The ReLu function and the cross-entropy loss function were selected as the activation functions of the hidden layer and readout layer respectively, and the weights were updated based on the back propagation algorithm. The audio classification used the same approach, while the difference is that the size of the network. This task only required a single-layer neural network, and the activation function of the output layer is also a cross-entropy loss function.

## Data availability

Source data for the figures are provided as a Source data file. All relevant data within the Supplementary Information are available from the corresponding authors upon request. Source data are provided with this paper.

## Code availability

All code used in simulations supporting this article is available from the corresponding authors upon request.

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

## Acknowledgements
This work was supported by the National Key R&D Program of China (No. 2019YFA0308500 to K.J.), the National Natural Science Foundation of China (No. 12222414 to C.G., No. 12074416 to C.G., No. 11721404 to K.J., No. 12174437 to C.W., No. 62075142 to Q.L.Z.), and the Youth Innovation Promotion Association of CAS (No. Y2022003 to C.G.).

## Author contributions
C.G. initiated the research. C.G. and K.J. supervised the project. P.L., G.L., and Z.L. prepared the sample. P.L., M.Z, and D.X. fabricated the device. The device measurements were done by P.L. with support from Z.W. Q.H.Z., and L.G. contributed to STEM measurements. Simulations were performed by M.Z. and D.X. P.L., M.Z., and C.G. wrote the manuscript. Q.L.Z, E.G., M.H., C.W., K.J., and G.Y. participated in the discussion of manuscript.

## Competing interests
The authors declare no competing interests.
