## [Peer Review File · Nature Communications]

Reviewers' comments:

Reviewer #1 (Remarks to the Author):

The authors have fabricated a neuromorphic electrolyte-gated transistor (EGT) by use of the BaSnO₃ (BSO) as the channel material. The BSO-EGT could perform physical reservoir and synaptic functions under optical and electrical stimuli. Moreover, the authors have investigated the multimodal characteristic of optoelectronic BSO-EGT, showing the potential on image recognition and dynamic gesture recognition. However, the following issues should be addressed before the publication of this manuscript.

1. The authors shows the long-term synaptic plasticity (long-term potentiation/depression) of an optoelectronic BSO-EGT in Figure 2h. However, the authors only use the potentiation in the EGT reservoirs. Could the authors comment on the use of the depression for the EGT reservoirs?
2. Figure 3d shows the initial and final current values of the reservoir state. It seems that the final currents of "0001" and "0010" are different. However, "0001" and "0010" is essentially the same. Could the authors tell the reason?
3. Figure 4f shows the recognition accuracy of the five information perception modes, indicating that mixed-signal modes are more universal. Could the authors explain the underlying mechanism of the difference between the single- and mixed-signal modes?
4. The BSO-EGT exhibits volatility and non-volatility under different external stimuli. The authors show the potential of volatility in a reservoir for image pre-processing. How about the application if the non-volatility of the BSO-EGT is considered?

Reviewer #2 (Remarks to the Author):

The authors propose an electrolytic gated device in BaSnO (BSO) with an ionic liquid DEME-TFSI as gate electrolyte that exhibits dynamics with tunable time-scales under optical and electrical stimuli. In particular, application of +1 V results in a volatile memory, whereas +2V results in a non-volatile memory. An optical signal applied before the electrical signal takes the device to a different state compared to optical or electrical alone (Supplementary Fig 15a).

The main questions for the authors are listed below:

- (ii) An actual system diagram or photograph highlighting the device structure with suspended gate electrode at a distance of 10 μm is missing both in diagram and description of methods. It must be quite challenging to achieve such a device, and to control both light and electrical signal selectively on a chip.
- (iii) The on/off ratios are extremely poor (<2), in fact all the states lie between 40-46, which may be the reason for the relatively poor learning efficiency. The off currents are high, so power consumption would be a problem. The ratio of currents in volatile to non-volatile devices is not remarkable (ie both very similar).
- (iv) The scan rates of the Current-Voltage characteristics are missing, under electrical, optical and combined stimulus.
- (v) There is a lack of physical/electrical evidence for their proposed mechanisms. For example the influence of D₂O in the IL is deemed to achieve non-volatility- this seems a very specific claim. Is this recoverable? Can the device be reset? Why should ion migration (heavy H) make the device non-

volatile ? The theory of oxygen vacancies and electrical double layer also seems surmised. They claim tunable ion dynamics with gate voltage in [35]. I refer them to Nanoionics-Based Three-Terminal Synaptic Device Using Zinc Oxide | ACS Applied Materials & Interfaces, which is a much simpler system compared to the present one. Could the authors highlight the advantages of their device and potential applications ?

(iv) Their dataset description is very confusing. It needs to be clarified which data was used in training and which was used in testing. How many samples were there in each category ? Is there a citation for the database ? What precautions were taken to avoid over-fitting ?

(v) The video classification section similarly can be improved in terms of readability. Which parts of the ANN were implemented in software and which in hardware is not easy to distinguish.

(iv) What is the meaning of pollution degree and tackle ? Please define.

(v) Very difficult to understand what is an audiovisual fusion task. How was this implemented ? How was the input converted into voltage and light signals ? The authors claim the improvement in combined efficiency for audio + video is due to multimode information processing. I am not convinced of that. Most articles report high efficiency for audio tasks due to limitations of the size of their dataset used in training, whereas for video the efficiencies reported are upto 94%. I fear the authors citations are incomplete on this front.

In summary, the methods of this article are not completely described. There needs to be more depth to their interpretation of the device mechanism and learning efficiency. There needs to be more substantive comparison with other state of the art implementations and context. What is the motivation for this type of a complex device with relatively less impressive performance ?

Reviewer #3 (Remarks to the Author):

Authors suggested an EGT-based synaptic transistor, utilizing BaSnO₃ films in an attempt to enhance the photoresponse characteristics of neuromorphic devices. However, there is some issue and lack of novelty to be published in "Nature communication" Therefore, my recommendation for this manuscript is "transfer to another journal in nature publication". The followings are my comments in detail for this manuscript.

Firstly, the EGT-based synaptic transistor presented in the study does not significantly differentiate itself from existing works in the field, particularly concerning the use of BaSnO₃ films. The lack of demonstrable superiority in aspects such as photoresponse and applicable light wavelengths underscores a notable absence of innovation.[1] Furthermore, the incorporation of both electrical and optical signals, though methodologically sound, follows a well-tread path, with experiments echoing conventional models like that of Pavlov's dog, thereby failing to exhibit marked progress or novel insights. [2]

The study does not explore the broader potential of neuromorphic devices beyond basic image transformation and data processing, notably omitting the exploration of high-resolution array configurations. [3] This limitation narrows the scope of the manuscript, diminishing its relevance and applicability in more advanced contexts that necessitate complex pattern recognition or extensive data processing capabilities.

The authors shows different decay characteristics under light and electrical stimuli, crucial for nuanced state representations in data processing. However, the manuscript lacks a detailed explanation of these

phenomena, impeding a thorough understanding of the device's intricate behavior. This gap is particularly concerning given the importance of these characteristics in practical applications.

[1] Wang, Xin, et al. "Enhanced Multiwavelength Response of Flexible Synaptic Transistors for Human Sunburned Skin Simulation and Neuromorphic Computation." *Advanced Materials* (2023): 2303699.

[2] Ke, Shuo, et al. "Artificial fear neural circuit based on noise triboelectric nanogenerator and photoelectronic neuromorphic transistor." *Applied Physics Letters* 123.12 (2023).

[3] Jo, Chanho, et al. "Retina-Inspired Color-Cognitive Learning via Chromatically Controllable Mixed Quantum Dot Synaptic Transistor Arrays." *Advanced Materials* 34.12 (2022): 2108979.

Manuscript ID: NCOMMS-23-45732-T

Title: Reconfigurable optoelectronic transistors for multimodal recognition

We thank all the reviewers for valuable comments regarding our research paper. Each of your insights have served to strengthen our manuscript. We have carefully revised the manuscript according to your constructive suggestions. Provided below is our detailed response to each comment raised.

Reviewer #1 (Remarks to the Author):

The authors have fabricated a neuromorphic electrolyte-gated transistor (EGT) by use of the BaSnO₃ (BSO) as the channel material. The BSO-EGT could perform physical reservoir and synaptic functions under optical and electrical stimuli. Moreover, the authors have investigated the multimodal characteristic of optoelectronic BSO-EGT, showing the potential on image recognition and dynamic gesture recognition. However, the following issues should be addressed before the publication of this manuscript.

Response:

Thank you very much for your hard-work in reviewing our manuscript. The point-by-point responses to the reviewer's comments are listed in the following.

Point 1.

The authors shows the long-term synaptic plasticity (long-term potentiation/depression) of an optoelectronic BSO-EGT in Figure 2h. However, the authors only use the potentiation in the EGT reservoirs. Could the authors comment on the use of the depression for the EGT reservoirs?

Response:

Thanks for the reviewer's valuable comment. Following your suggestions, we will elaborate the feasibility of using the depression for the EGT reservoirs.

First of all, we agree with your opinion that the depression behavior can indeed be used to implement reservoir computing (RC), as long as the behavior has nonlinear and volatile characteristics^{R1}.

Here, the long-term synaptic plasticity (long-term potentiation/depression) of BSO-EGT in Fig. 2h is used to realize synaptic function in artificial neural networks, whereas the reservoir is a functional element that exploits nonlinear relaxation of short-term synaptic plasticity (short-term potentiation/depression) for pre-processing^{R2}. Therefore, as long as the relaxation process of the device has nonlinear and volatile characteristics, in theory, whether the potentiation or depression curves can be used to implement RC.

However, our as-prepared BSO-EGT is in a high-resistance state, and the downward adjustment in the pristine state would cause significant leakage, making it difficult to

perform RC based on the depression process. This can be easily observed from the transfer curve (Supplementary Fig. 9). In comparison, it is easier and more stable to increase the conductance and current of the device, and its relaxation process is also easier to observe and sampling. So, we chose the potentiation of BSO-EGT to perform RC.

Although the present scheme cannot utilize the depression behavior to implemented the reservoir, there have been recent studies that have realized this function^{R3}. For example, Liu *et al.* proposed an interface-type dynamic transistor gated by an $\text{Hf}_{0.5}\text{Zr}_{0.5}\text{O}_2$ (HZO) film to perform reservoir computing^{R4}. The channel conductance of Mott material $\text{La}_{0.67}\text{Sr}_{0.33}\text{MnO}_3$ (LSMO) can effectively be modulated by taking advantage of the coupled property of the polarization process and oxygen migration in hafnium-based ferroelectrics. The application of positive voltages would decrease the channel conductance, and the decay in current after the removal of the pulse fits a double-exponential function well. Therefore, this nonlinear relaxation behavior in the depression process can be used for application in RC, capable of performing wave classification and chaotic prediction tasks.

Corresponding references

R1. Wu XS, *et al.* Wearable in-sensor reservoir computing using optoelectronic polymers with through-space charge-transport characteristics for multi-task learning. *Nature Communications* 14, 468 (2023).

R2. Moon J, *et al.* Temporal data classification and forecasting using a memristor-based reservoir computing system. *Nature Electronics* 2, 480-487 (2019).

R3. Chen ZW, *et al.* All-ferroelectric implementation of reservoir computing. *Nature Communications* 14, 3585 (2023).

R4. Liu Z, *et al.* Interface-type tunable oxygen ion dynamics for physical reservoir computing. *Nature Communications* 14, 7176 (2023).

Changes made:

We added “It should be noted that as long as the relaxation process of the device has nonlinear and volatile characteristics, in theory, whether the potentiation or depression curves can be used to implement RC. Our as-prepared BSO-EGT is in a high-resistance state, the downward adjustment in the pristine state will cause significant leakage, making it difficult to perform RC based on the depression process. But there have been recent studies utilizing the depression behavior to implement reservoirs^{52, 53}” [Paragraph 1, page 10]

Corresponding references were added in the revised manuscript.

52. Chen ZW, *et al.* All-ferroelectric implementation of reservoir computing. *Nature Communications* 14, 3585 (2023).

53. Liu Z, *et al.* Interface-type tunable oxygen ion dynamics for physical reservoir computing. *Nature Communications* 14, 7176 (2023).

Point 2.

Figure 3d shows the initial and final current values of the reservoir state. It seems that the final currents of “0001” and “0010” are different. However, “0001” and “0010” is essentially the same. Could the authors tell the reason?

Response:

Thanks for your valuable comments. In order to better explain the question raised by the reviewer, the different behaviors of devices that realize reservoir function and synaptic function after external stimulation are listed below.

First, we would explain the encoding approach of external stimuli for better understanding. This approach is not unique in RC and has been used in many other reports as well^{R5-R7}. We are dealing with time-related dynamic tasks, so the final result should contain time information. “0” and “1” represent the “off” and “on” states of external stimulations applied to the device, respectively. In our study, the external stimulations can be either optical or electrical pulse, and the time interval between each two states is 1 s. The current of the device in the first second after the end of the four continuous states is collected and used for subsequent processing. Taking “0001” as an example, the device samples the first second after the stimulus “1” is applied. And “0010” is sampled at the first second after the last “0” is applied. Although our sampling rules are the same, in these two cases, the current of the device undergoes relaxation of different time lengths after stimulation. For a clearer description, please refer to Fig. R1.

Fig. R1. Comparison of reservoir and synaptic properties

Devices that implement RC or synapse functions have different characteristics. For synaptic devices, since they can maintain a non-volatile state after receiving external stimulation^{R8}, the final state of the synaptic device depends on the number of pulses received. In this case, regardless of the order in which the stimuli are applied, the device will operate in the same state when the same number of stimuli are received. But this limits the application of synaptic devices in tasks involving temporal information. “0001” and “0010” are exactly the same state in the synaptic devices.

Reservoirs have received widespread attention because they can handle sequence information containing temporal correlations. Implementing the reservoir functionality requires that the current state of the device can be affected by historical stimuli^{R9}. Different combinations of stimuli will produce different states, allowing sequential information processing, which relies on the nonlinear relaxation process after being stimulated. Therefore, “0001” and “0010” are not the same two sequences for reservoirs.

Corresponding references

R5. Sun L, *et al.* In-sensor reservoir computing for language learning via two-dimensional memristors. *Science Advance* 7, eabg1455 (2021).

R6. Zhang Z, *et al.* In-sensor reservoir computing system for latent fingerprint recognition with deep ultraviolet photo-synapses and memristor array. *Nature Communications* 13, 6590 (2022).

R7. Yang JY, *et al.* Reconfigurable Physical Reservoir in GaN/ α -In₂Se₃ HEMTs Enabled by Out-of-Plane Local Polarization of Ferroelectric 2D Layer. *Acs Nano* 17, 7695-7704 (2023).

R8. Li G, *et al.* Photo-induced non-volatile VO₂ phase transition for neuromorphic ultraviolet sensors. *Nature Communications* 13, 1729 (2022).

R9. Tan H, van Dijken S. Dynamic machine vision with retinomorphic photomemristor-reservoir computing. *Nature Communications* 14, 2169 (2023).

Changes made:

We added “Due to the nonlinear relaxation characteristics of the reservoir, its final state depends on its activity history. Therefore, “0001” and “0010” are two different sequences for the reservoir.” [Paragraph 1, page 10]

Point 3.

Figure 4f shows the recognition accuracy of the five information perception modes, indicating that mixed-signal modes are more universal. Could the authors explain the underlying mechanism of the difference between the single- and mixed-signal modes?

Response:

Thanks for your valuable comment. According to your question, we would elaborate on the difference between the single- and mixed-signal modes.

When humans obtain rich, diverse and complex external information, they perceive visual signals through the eyes, auditory information through the ears, and tactile information through the trunk. Multi-channel information can complement and verify each other to maximize the accuracy of the final decision. In recent years, the development of robotic systems and autonomous driving has placed higher requirements on the robustness of object recognition in complex environments^{R10, R11}. Most of the proposed methods rely on vision. However, the processing of visual tasks is seriously affected by the brightness of the surrounding environment and the presence of occlusions. Therefore, the study of multi-sensory integration in simulated organisms has attracted much attention. Progress has been made in integrating visual information with other sensory abilities such as somatosensory and hearing^{R12, R13}.

In Figure 4, the powerful learning ability of the RC is used to recognize polluted images containing multiple types of information. In these images, we defined that the information of unpolluted part can obtain through vision, while the other part only can be obtained through touch. With tactile sensors, touch information can be converted into electrical information. For the convenience of demonstration, we directly applied the converted electrical signal to our transistor device, while the observable part is gathered by optical signals. When just one channel is used to obtain information, only parts of the features can be obtained, and the recognition accuracy is lower. However,

the mixed mode can collect more comprehensive and rich information, so a higher recognition accuracy is obtained^{R14, R15}. We simulated the process of using multiple senses to achieve high levels of cognition. This is similar to multi-sensory perception and fusion processing in humans.

Corresponding references

R10. Li G, *et al.* Skin-inspired quadruple tactile sensors integrated on a robot hand enable object recognition. *Science Robotics* 5, eabc8134 (2020).

R11. Liu M, *et al.* A star-nose-like tactile-olfactory bionic sensing array for robust object recognition in non-visual environments. *Nature Communications* 13, 79 (2022).

R12. Wang M, *et al.* Gesture recognition using a bioinspired learning architecture that integrates visual data with somatosensory data from stretchable sensors. *Nature Electronics* 3, 563-570 (2020).

R13. Chen A, *et al.* Multi-information fusion neural networks for arrhythmia automatic detection. *Computer Methods Programs Biomed* 193, 105479 (2020).

R14. Liu K, *et al.* An optoelectronic synapse based on α -In₂Se₃ with controllable temporal dynamics for multimode and multiscale reservoir computing. *Nature Electronics* 5, 761-773 (2022).

R15. Jiang C, *et al.* Mammalian-brain-inspired neuromorphic motion-cognition nerve achieves cross-modal perceptual enhancement. *Nature Communications* 14, 1344 (2023).

Changes made:

We added “When just one channel is used to obtain information, only parts of the features can be obtained, and the recognition accuracy is low. However, the mixed mode can collect more comprehensive and rich information, so a higher recognition accuracy can be obtained.” [Paragraph 1, page 12]

Point 4.

The BSO-EGT exhibits volatility and non-volatility under different external stimuli. The authors show the potential of volatility in a reservoir for image pre-processing. How about the application if the non-volatility of the BSO-EGT is considered?

Response:

Thanks for the reviewer’s valuable suggestion. In this work, both volatile and non-volatile properties exhibited by the reconfigurable device are utilized. Briefly, we use volatile property for the reservoir and non-volatile property for artificial synapse functionality, respectively.

The reservoirs have the ability to process sequence data, which can not only integrate time series information to compress the data volume, but also map input data to high-dimensional space for preprocessing to facilitate subsequent classification^{R16, R17}. However, during the operation of RC, it is still necessary to use specialized device array as the readout layer to storage and update weights for classification tasks^{R18, R19}, where

artificial synapses play an important role. This makes the devices within the readout layer should have good non-volatility.

In this work, we demonstrated that our devices are capable of exhibiting both volatile and nonvolatile properties upon external stimulation. Then, we used volatile property for reservoir function and non-volatile property for readout layer weight updates. Finally, we realized the dynamic information recognition task based on the reconfigurable BSO-EGTs.

Corresponding references

R16. Wu X, *et al.* Wearable in-sensor reservoir computing using optoelectronic polymers with through-space charge-transport characteristics for multi-task learning. *Nature Communications* 14, 468 (2023).

R17. Sun L, *et al.* In-sensor reservoir computing for language learning via two-dimensional memristors. *Science Advance* 7, eabg1455 (2021).

R18. John RA, *et al.* Reconfigurable halide perovskite nanocrystal memristors for neuromorphic computing. *Nature Communications* 13, 2074 (2022).

R19. Wang S, *et al.* An organic electrochemical transistor for multi-modal sensing, memory and processing. *Nature Electronics* 6, 281-291 (2023).

Changes made:

To clearly show the application of non-volatile characteristic, we changed “A reservoir needs nonlinear dynamics to transform low-dimensional signals into a high-dimensional state space¹⁸⁻²⁰, which can improve the computing efficiency to reduce time and energy consumption^{21, 22}. The hardware implementation of reservoirs should have volatility, so that the current state of a device is related to its previous experience but not influenced by the distant past. Then, pre-processed signals are transmitted to an artificial neural network (ANN) for information integration and inference. Artificial synaptic devices can simulate the plasticity of biological ones, and in the hardware implementation of ANNs they are responsible for post-processing and storage, which requires non-volatility.” to “A reservoir relies on nonlinear dynamics to convert low-dimensional signals into a high-dimensional state space¹⁸⁻²⁰, thereby enhancing computing efficiency and reducing time and energy consumption^{21, 22}. The hardware implementation of reservoirs should exhibit volatility, ensuring that the current state of a device can be influenced by its recent experience without being affected by distant past events. Subsequently, pre-processed signals are conveyed to an artificial neural network (ANN) for information integration and inference. Artificial synaptic devices can replicate the plasticity of biological synapses, and in the hardware implementation of ANNs, they are responsible for post-processing and storage, necessitating non-volatility²³⁻²⁵.” [Paragraph 2, Page 3]

We changed “In addition, BSO-EGT exhibit volatility and non-volatility under different external stimuli. The volatile characteristic combined with the nonlinearity of the device can be used as a reservoir for image pre-processing. The non-volatile property, on the other hand, meets the needs of artificial synapses, as it enables post-processing

and storage functions^{16, 32.}” to “Furthermore, BSO-EGTs demonstrate both volatility and non-volatility in response to various external stimuli. The volatile nature, coupled with the nonlinearity of the device, offers great potential as a reservoir for image pre-processing. In contrast, the non-volatile property fulfills the requirements of artificial synapses, as it facilitates post-processing and storage functionalities^{16, 30.}” [Paragraph 1, page 11]

Reviewer #2 (Remarks to the Author):

The authors propose an electrolytic gated device in BaSnO₃ (BSO) with an ionic liquid DEME-TFSI as gate electrolyte that exhibits dynamics with tunable time-scales under optical and electrical stimuli. In particular, application of +1 V results in a volatile memory, whereas +2V results in a non-volatile memory. An optical signal applied before the electrical signal takes the device to a different state compared to optical or electrical alone (Supplementary Fig 15a).

Response:

We greatly thank the reviewer for the valuable comments. We have improved the manuscript by taking into account the comments/suggestions from the reviewer. Please find our responses to the comments below.

Point 1.

An actual system diagram or photograph highlighting the device structure with suspended gate electrode at a distance of 10 μm is missing both in diagram and description of methods. It must be quite challenging to achieve such a device, and to control both light and electrical signal selectively on a chip.

Response:

Thanks for the reviewer's comment. We apologize for not showing optical photos of the devices in the original manuscript, which sent a vague message to the reviewers. In this work, we used a coplanar (side-gate) transistor structure instead of a suspended gate. The gate, source and drain electrodes were deposited on the MgO substrate through a magnetron sputtering process, and the distance between the gate and the channel was 10 μm . This structure is conducive to both electrical and optical modulation of the channel conductance, and the manufacturing process is simple. Based on the concern raised by the reviewer, we have modified the corresponding description in the methods section and added an optical image of the device in the Supplementary Information.

Fig. R2. Optical microscopy image of a coplanar (side-gate) BSO-EGT (left) and the left view of the device structure (right).

Changes made:

To elaborate the structure of the BSO-EGT, we changed “The thin films were patterned into channels with a coplanar gate structure using standard photolithography and argon-ion etching. The effective device area is $50\ \mu\text{m}\times 180\ \mu\text{m}$. The length between the gate electrode and channel was $10\ \mu\text{m}$. Pt layer with $30\ \text{nm}$ thickness was deposited as electrodes by RF sputtering.” to “Through photolithography and ion beam etching technology, the BSO film is patterned into channels with an effective area of $50\times 180\ \mu\text{m}^2$. A coplanar side-gate structure is adopted, and a $30\ \text{nm}$ Pt layer is deposited as an electrode by magnetron sputtering. The distance between the gate and the channel is $10\ \mu\text{m}$.” [Method, Device fabrication]

We added optical microscopy image and side view of the device in the Supplementary Figure 4.

Supplementary Figure 4. Optical microscopy image of a coplanar (side-gate) BSO-EGT (left) and the left view of the device structure (right).

We added “Supplementary Figure 4 shows the optical microscopy image of the coplanar side-gate BSO-EGT and the left view of the device structure diagram.” [Paragraph 2, page 5]

We also modified the serial numbers of subsequent Supplementary Figures in the revised manuscript and supplementary information.

Point 2.

The on/off ratios are extremely poor (<2), in fact all the states lie between 40-46, which may be the reason for the relatively poor learning efficiency. The off currents are high, so power consumption would be a problem. The ratio of currents in volatile to non-volatile devices is not remarkable (ie both very similar).

Response:

Thanks to the reviewer for the valuable comments. Based on your suggestions, we prepared a batch of BSO samples under different growth conditions to further improve the device performance. Finally, the improved on/off ratio of the transfer curve was increased to 10^6 (Figure R3a). Under illumination, the light/dark current ratio increased

to more than an order of magnitude. The off-state current is now reduced from 40 nA to 0.1 nA, and the single spike energy consumption is 12 pJ (Figure R3b).

We found that increasing the oxygen pressure during the film growth will lead to a decrease in the film conductivity. It is attributed to the increase in the oxygen vacancy content^{R20, R21}. This can be attributed to that ionized oxygen vacancies could provide the additional mobile carriers for conduction, leading to the increased carrier density, and oxygen vacancies can neutralize the negative charges at threading dislocations, which suppresses the contribution of dislocation scattering^{R21}. The off-state current of newly prepared device is 0.1 nA, which can reduce power consumption. When applying a 20 ms pulse voltage stimulus (1 V), which is of the same duration as the work recommended from the reviewer, the power consumption of a single spike is 12 pJ (Figure R3b). It should be noted that this value is lower than the reviewer's recommended article (~ 35 pJ). Meanwhile, the on/off ratio of the transfer curve was increased to 10^6 (Figure R3a).

Fig. R3. **a**, Transfer curves of the BaSnO₃ transistor measured with $V_{SD} = 0.5$ V. **b**, Power consumption of a single spike when a 20 ms pulse voltage stimulus (1 V) is applied. **c**, Evolution of channel current under pulse voltage stimulation of $V_G = 1$ V and $V_G = 3$ V. **d**, The non-volatile multi-level conductance retention properties. The multi-states are produced by a series of V_G pulses with different durations ($V_G = +2.5$ V, durations from 1 s to 5 s).

The problem that the difference between volatile and non-volatile currents is not obvious enough mentioned by the reviewer has also been improved. It can be seen that under 1 V voltage stimulation, the channel current quickly returns to the initial state, however, under 3 V voltage stimulation, the current change shows obvious non-volatility (Figure R3c). Longer voltage pulse stimulation was also examined. When stimulated by a 1 V voltage pulse, the device still showed obvious volatile characteristics as the pulse time increased. Only when the applied voltage exceeds the hydrolysis voltage threshold does the non-volatile property appear (Figure R4f). Under this condition, as the pulse time increases, the non-volatile characteristics become more obvious. We further examined the state programmability feature. It was found that its conductance change did not show obvious decay within 300 s (Figure R3d). Therefore, distinct volatile and non-volatile characteristics can be obtained in the revised manuscript. We would like to stress that the volatile and nonvolatile characteristics of conductance change are only determined by the voltage amplitude. This is very important in accurately achieving volatile and non-volatile switching.

Next, we found that due to the low initial oxygen vacancy content of the film, a greater proportion of oxygen vacancies can be generated by UV irradiation. Therefore, the conductance change is more obvious under UV illumination, showing an order of magnitude enhancement. Therefore, a larger light/dark current ratio is demonstrated. The reviewer mentioned that the low learning efficiency may be caused by the small conductance variation range of the device. This problem was solved by the newly fabricated device showing a larger conductance variation range. We retested all experimental data and updated the corresponding figures based on the new device in the revised manuscript (Fig. R4), so the reviewer's concerns on switch ratio, energy consumption and learning efficiency were addressed by supplementing experiments.

Fig. R4. EGT with reconfigurable characteristics for multimode sensing. **a**, Schematic illustration of the neuromorphic transistor that can be stimulated using optical and electrical signals. The BaSnO₃ film serves as a channel between the source (S) and drain (D) electrodes, and IL is used as the gating medium. **b**, Short-term potential effect stimulated using optical pulses (1s) with a 0.5 V bias to monitor the channel current. **c**, Transition from STM to LTM under different light durations. **d**, Memory decay process after light pulse (70 mW/cm² for 1 s) stimulation, fitted using a double-exponential function. The fitting result shows the coexistence of two physical processes. **e**, Excitatory post synaptic current (PSC) induced via a train of optical pulse (70 mW/cm², 1s with intervals of 1, 5, 20 and 30 s) with a 0.5 V bias to monitor the channel current. **f**, The channel current controlled through a series of V_G pulse with a pulse width of 1s and different amplitudes which shows the transition of the PSC from volatile to non-volatile property. **g**, Spike number-dependent plasticity under electrical stimuli (+2 V, 1s) with the same intervals of 1s. **h**, Cyclic controlled LTP using V_G (equally spaced from +1.5 to +3.5 V, duration of 1s, spaced 1s apart) and LTD using V_G (equally spaced from -1.5 to -3.5 V, duration of 1s, spaced 1s apart) for 32 pulses.

Corresponding references

R20. Luo BC, Hu JB. Unraveling the Oxygen Effect on the Properties of Sputtered BaSnO₃ Thin Films. *ACS Applied Electronic Materials* 1, 51-57 (2019).

R21. Luo B, Hu J. Unraveling the Oxygen Effect on the Properties of Sputtered BaSnO₃ Thin Films. *ACS Applied Electronic Materials* 1, 51-57 (2018).

R22. Yoon D, Yu S, Son J. Oxygen vacancy-assisted recovery process for increasing electron mobility in n-type BaSnO₃ epitaxial thin films. *Npg Asia Materials* 10, 363-371 (2018).

Changes made:

We changed “The 10 nm BaSnO₃ film was epitaxially grown on (001)-oriented MgO substrates at 780 °C under O₂ pressure of 1.5 Pa.” to “The 10 nm BaSnO₃ film was epitaxially grown on (001)-oriented MgO substrates at 780 °C under O₂ pressure of 5.5 Pa.” [Method, Sample preparation]

We changed “However, as the light intensity gradually increased, the optical response of the device changed from volatility to non-volatility, similar to the short-term memory (STM) to long-term memory (LTM) transition of biological synapses (Fig. 2c).” to “However, as the duration gradually increased, the optical response of the device changed from volatility to non-volatility, similar to the short-term memory (STM) to long-term memory (LTM) transition of biological synapses (Fig. 2c).” [Paragraph 3, page 5]

We added “Supplementary Figure 10 shows the transistor characteristic curves of BSO-EGT, namely the transfer characteristic curve and the output characteristic curve.” [Paragraph 2, page 7]

We added “Thus, “LLLE”, “LLEE” and “LEEE” can complete image recognition tasks with around 90% recognition accuracy at all contamination levels.” [Paragraph 1, page 12]

We have updated all of the corresponding figures with the newly fabricated device.

Fig. 2 | EGT with reconfigurable characteristics for multimode sensing. **a**, Schematic illustration of the neuromorphic transistor that can be stimulated using optical and electrical signals. The BaSnO₃ film serves as a channel between the source (S) and drain (D) electrodes, and IL is used as the gating medium. **b**, Short-term potential effect stimulated using optical pulses (1s) with a 0.5 V bias to monitor the channel current. **c**, Transition from STM to LTM under different light durations. **d**, Memory decay process after light pulse (70 mW/cm² for 1 s) stimulation, fitted using a double-exponential function. The fitting result shows the coexistence of two physical processes. **e**, Excitatory post synaptic current (PSC) induced via a train of optical pulse (70 mW/cm², 1s with intervals of 1, 5, 20 and 30s) with a 0.5 V bias to monitor the channel current. **f**, The channel current controlled through a series of V_G pulse with a pulse width of 1s and different amplitudes which shows the transition of the PSC from volatile to non-volatile property. **g**, Spike number-dependent plasticity under electrical stimuli (+2 V, 1s) with the same intervals of 1s. **h**, Cyclic controlled LTP using V_G (equally spaced from +1.5 to +3.5 V, duration of 1s, spaced 1s apart) and LTD using V_G (equally spaced from -1.5 to -3.5 V, duration of 1s, spaced 1s apart) for 32 pulses.

Fig. 3| Nonlinear mapping of multimodal signals based on EGT reservoirs. **a**, Combination mode for 16 inputs of 4-bit optical and electrical pulses. Here, both the optical and electrical pulse widths were 1 s and the pulse interval was 1 s. **b**, Response characteristics of the channel current in 1111, 1110, 1100, 1000 combinations under the LLE mode input (light intensity 70 mW/cm², V_G = 1 V). **c**, The distinguishable output of 4-bit reservoir states reading at 0.3 V. **d**, Initial and final current values of the reservoir state with pulse stimulation of 16 combinations in the LLE inputs mode.

Fig. 4 Fused information input reservoir for recognition of multimodal Fashion-MNIST datasets. **a**, Image perception in complex environments. The right half of information can only be perceived through electrical signals, while the left half of information can only be perceived through visual signals. The original image ($28 \times 28 = 764$ pixels) was reorganized into a 196×4 by column vector array. **b**, Output results of the reservoir in the “LLLL” sensing mode, when 10% (top) and 90% (bottom) of the image can only be read by electrical signals. **c**, The change of ANN synaptic weight before and after training. The trained synaptic weight conforms to the normal distribution. **d**, **e**, Recognition accuracy under the five perception methods when 0% and 50% of the image can only be collected by electrical signals, **f**, Recognition accuracy of the five information perception modes variation with the proportion of the contaminated part (which can only be read via electrical signals).

Fig. 5 Mimics human perception of dynamic audiovisual information. **a**, Simplified diagram of humans achieving high levels of cognition through audiovisual integration. **b**, Three-dimensional space mapping of “Rolling hand toward left” gesture, a sample from the EgoGesture dataset. **c**, Images superimposed at four time points in the intermediate state after hand movements were preprocessed by the reservoir. **d**, Speech spectrogram of “Rolling hand toward left” gesture, reflecting the frequency domain distribution of audio at different moments. Corresponding voice overs for gestures, generated using text-to-speech engine. **e**, Input vector after normalization of time and frequency domain information, and output vector after reservoir processing. **f**, Comparison of recognition accuracy under single- and multi-mode information processing.

Supplementary Figure 5. The effect of UV light intensity on the channel current. Relaxation time of BaSnO₃ device under various light intensity with the same illumination durations 1s.

Supplementary Figure 6. Paired pulse facilitation (PPF) plasticity. **a**, Channel current is measured under UV irradiation with identical light intensity and duration time (light intensity of 70 mW/cm², the duration of 1 s). **b**, PPF ratio as a function of the pulse intervals, where the red line represents fitting results using the double exponential decay function.

Supplementary Figure 7. Influence of pulse time interval on channel current. **a**, Changes of channel current stimulated by UV pulses (light intensity of 70 mW/cm^2 , the duration of 1 s) with varying pulse intervals. **b**, Relationship between the channel current and the number of pulses, the current value after the pulse is applied for 1 s is used as the sampling point

Supplementary Figure 8. The effect of the light exposure on the channel current under blue laser irradiation. **a**, Device response to illumination of different wavelengths (70 mW/cm^2 , duration 1 s) under the same conditions. Among them, UV causes a more obvious change in conductance. **b**, ISD response to blue light irradiation

at different durations (140 mW/cm^2). **c**, Spike number dependent plasticity under blue light irradiation with the same pulse width and intervals (140 mW/cm^2 for 1 s). **d**, Relationship between channel current and sequence pulse interval, the channel current is measured under light irradiation with different interval duration (light intensity of 140 mW/cm^2 , the duration of 1 s)

Supplementary Figure 11. Electrical performance of BaSnO₃ transistor. **a**, Transfer curve of the BaSnO₃ electrolyte-gated transistor (BSO-EGT) measured on $V_{SD}=0.5 \text{ V}$. The gate voltage was swept from 0 V to 2.5 V, 2.5 V to -2.5 V, and then back to 0 V. **b**, Output curve of BSO-EGT at a V_{DS} range of 0 to 1.5 V.

Supplementary Figure 12. Long-term and short-term plasticity of BaSnO₃ transistors. **a**, Under the voltage stimulation of $V_G = 1 \text{ V}$ (with different durations), the channel current can quickly return to the initial state. **b**, Under the electrical pulse stimulation of $V_G = 2 \text{ V}$, the conductance of the device exhibits obvious non-volatile characteristics.

Supplementary Figure 13. Spike-frequency-dependent plasticity. Current change subject to 8 voltage pulses ($V_G = +1$ V, 1 s) with different pulse intervals.

Supplementary Figure 14. The non-volatile multi-level conductance retention properties. The multi-states are produced by a series of V_G pulses with different durations ($V_G = +3$ V, durations from 1 s to 5 s).

Supplementary Figure 15. Pulse-switching characteristics. The high conductance state is generated by $V_G = 2$ V, and the low conductance state is generated by $V_G = -2$ V. The pulse duration is 1 s. We choose 1 s after applying the voltage pulse stimulation as the sampling point.

Supplementary Figure 17. Device Characteristics for Mixed-Mode Input. **a**, Changes in channel current are monitored by a 0.5 V read voltage at the three mode inputs (“LL”, “LE”, “EE”). Among them, “LL” represents two optical pulses (70 mW/cm²), “LE” represents one optical pulse and one electric pulse, and “EE” represents two electric pulse signal inputs (1 V). The pulse width and time interval are both 1 s. **b**, Under “LE” mode input, the influence of the time interval between two pulses on the current value of the sampling point, the current value is collected 1 s after the second pulse is applied.

Supplementary Figure 18. Nonlinear mapping of 4-bit inputs based on the BSO reservoir. **a, b,** Changes in channel currents in response to light and electrical stimulation with different coding combinations, respectively. **c, d,** show the final distinguishability of the 16 encodings under light and electrical stimulation, respectively. The final distinguishable state is taken at 1 s after the application of the fourth pulse

Supplementary Figure 19. Multimodal nonlinear dynamics for reservoir computing. Variety of input waveform patterns. **a**, “LEEE” mode, **b**, “LLEE” mode demonstrate different current states distribution. **c**, The distinguishable output of 4-bit reservoir states of “LEEE” input. **d**, The distinguishable output of 4-bit reservoir states of “LLEE” input.

Point 3.

The scan rates of the Current-Voltage characteristics are missing, under electrical, optical and combined stimulus.

Response:

Thanks for the reviewer’s valuable comments on our manuscript. The scan rate during the transfer characteristic curve test is 10 mV/s. While other experiments were analyzed based on current-time curves.

During optical stimulus, the scanning rate is 9 sampling points/second. During electrical stimulus, the scanning rate is 5 sampling points/second. During combined stimulus, the scanning rate is 9 sampling points/second. And the scan rate of transfer characteristic curve is 10 mV/s

Changes made:

To supplement the sampling rate of the experimental data, we changed “The channel current I_{SD} of the device was measured under different UV exposure conditions.” to

“The channel current I_{SD} of the device was measured under different UV exposure conditions, and the scanning rate is 9 sampling points/second.” [Paragraph 3, page 5]

We changed “Supplementary Figure 10 illustrates the transfer curves measured along the counterclockwise direction, showing a typical transistor characteristic. The response of the device to electrical pulse stimulation was then measured at $V_{SD} = +0.3$ V.” to “Supplementary Figure 10a illustrates the transfer curves measured along the counterclockwise direction, and the scan rate is 10 mV/s. The response of the device to electrical pulse stimulation was then measured at $V_{SD} = 0.3$ V.” [Paragraph 2, page 7]

We added “During electrical stimulus, the scanning rate is 5 sampling points/second.” [Paragraph 2, page 7]

We added “And the scanning rate is 9 sampling points/second.” [Paragraph 2, page 9]

Point 4.

There is a lack of physical/electrical evidence for their proposed mechanisms. For example the influence of D_2O in the IL is deemed to achieve non-volatility- this seems a very specific claim. Is this recoverable? Can the device be reset? Why should ion migration (heavy H) make the device non-volatile? The theory of oxygen vacancies and electrical double layer also seems surmised. They claim tunable ion dynamics with gate voltage in [35]. I refer them to Nanoionics-Based Three-Terminal Synaptic Device Using Zinc Oxide | ACS Applied Materials & Interfaces, which is a much simpler system compared to the present one. Could the authors highlight the advantages of their device and potential applications?

Response:

Thanks for the reviewer’s valuable comments on our manuscript. We apologized for the lack of clarity on our description of the physical mechanisms. Based on the reviewer’s questions, we would supplement the corresponding experiments, and discuss the underlying mechanisms of the different behaviors of BSO-EGT under optical and electrical stimulation.

1. The volatile response of BSO-EGT under electrical stimulation originated from the electrical double layer (EDL). In this mechanism, ionic liquids are conductors of ions but are electrically classified as insulators. The rapid movement of anions and cations under an electric field produces a strong accumulation of space charge at the interface between the electrolyte and the channel, which exists in the form of an EDL, called a Helmholtz layer^{R23}. In theory established in 1853, an EDL can be viewed as a simple parallel plate capacitor. When a positive bias is applied to the gate, the anions in the ionic liquid move toward the interface between the electrolyte and the gate, due to the effect of the electric field^{R24}. At the same time, correspondingly, the cations in the ionic liquid migrate to the interface between the electrolyte and the channel to form a

Helmholtz layer. Accumulation occurs due to blocking ions in the solid channel. In order to balance the EDL formed at the interface, an accumulation of electrons occurs inside the channel. Therefore, the conductance of the channel will change. When the external bias voltage is removed, the ions at the interface spontaneously migrate back into the ionic liquid, and the electrical double layer disappears, so the channel conductance returns to its original state.

2. First of all, we need to explain that the non-volatility of the device is not caused by D_2O , which just acts as an isotope marker to verify the source of the protons (H^+).

The conductance of the BSO channel showed non-volatile changes under voltage stimulation, which was confirmed by experimental results (Figure R5a, b). The theoretical value of the electrolysis voltage of pure water is 1.23 V^{R25}. Considering the influence of experimental conditions, it will generally be higher than this value^{R26}. We found that this value is approximately 1.3 V in our experiments. Because at this voltage, there is a peak in the gate current of the transfer characteristic curve (Figure R5a), which is related to the hydrolysis reaction^{R27}.

Fig R5. **a**, Transfer curve of the $BaSnO_3$ electrolyte-gated transistor (BSO-EGT) measured on $V_{SD}=0.5$ V. **b**, Transfer curve for 15 consecutive scans between $V_G = -3V \sim +2.5V$. **c**, The channel current controlled through a series of V_G pulse with a pulse width of 1s and different amplitudes which shows the transition of the PSC from volatile to non-volatile property.

Below the threshold voltage, the device exhibits a volatile phenomenon due to the formation of EDL near the interface^{R28}, while under stimulation above the critical voltage, the channel conductance exhibits an obvious non-volatile phenomenon (Fig. R5c). This is because, in addition to the presence of EDL at the interface, H^+ originating from the trace water containing in the ionic liquids would be injected into the film when the positive voltage exceeded the critical voltage. Protons (H^+) produced by hydrolysis can be driven to the channel interior, resulting in strong interactions with the solid material^{R23, R27}. Therefore, the non-volatility of the device came from the migration of protons generated by hydrolysis in the ionic liquid into the oxide film under the positive gating, causing a non-volatile increase in the channel conductance.

To verify this mechanism, we performed secondary ion mass spectrometry (SIMS) experiments. The results (Fig. R6a) show that hydrogen ions appear inside the electrically modulated films, and the ion concentration rises significantly as the voltage increases. Moreover, the addition of a small volume of D₂O to the ionic liquids resulted in the presence of D⁺ signal in the electrically-modulated film, which was not observed inside the film without modulation (Fig. R6b). This result clearly demonstrated that the inserted hydrogen ions under positive gating come from the trace water inside ionic liquids.

Fig R6. a, b Secondary-ion mass spectrometry (SIMS) depth profiling in the pristine and gated BaSnO₃ films.

The non-volatile change in conductivity caused by ion migration (H⁺, Li⁺) is not a unique conclusion, and has been reported in organic^{R29} and inorganic materials^{R30}. Since protons have a smaller ionic radius than other ions, they are more conducive to migrate into the interior of the film^{R27}. The above mechanism has been demonstrated in many electrochemical transistors.

3. The reversibility of the non-volatile behavior is also demonstrated in the revised manuscript. The transfer characteristic curve of the device was continuously measured 15 times, (Fig. R5b) and there was no obvious degradation in device performance. The conductance increase caused by ion migration can be restored to the initial state under the action of negative voltage, so the curves overlap well. When a positive pulse above the threshold voltage was applied to the device, a significant non-volatile increase in channel conductance occurs. Using negative gate voltage pulse stimulation, this process can proceed in the reverse direction. This was well verified during the LTP and LTD measurements (Fig. 2h), where we made the device reach a high conductance state by applying positive voltage stimulation and decreased the conductance by corresponding negative pulses. In the transfer characteristic curve, it can be seen that when a cycle scan ends, the conductance of the device returns to near the initial state. In brief, all the above experiments confirm that the device can be reset.

4. Previous studies have verified that oxygen vacancies will be generated in BSO films under ultraviolet (UV) irradiation, resulting in an increase of film conductance. Lee *et al.* used ambient-pressure X-ray photoemission spectroscopy (APXPS) for in-situ characterization (Figure R6) to monitor the chemical origin of generated defects induced by UV irradiation in real time^{R31}. O 1s spectra of BaSnO₃ epitaxial films were acquired, with UV-light illumination under vacuum and without illumination in oxygen atmosphere. After the UV light was illuminated on the BaSnO₃, the binding energy of O 1s shifted toward higher with development of an additional peak shoulder that strengthened as illumination duration increased. Before the UV illumination under vacuum, the O 1s spectra could be mostly assigned to the peak related to lattice oxygen (O(I) at 529.9 eV) with a tiny fraction of the peak related to defective oxygen (O(II) at 531.3 eV)^{R32}. The peak shifts toward higher binding energy upon the illumination of UV light, the area fraction of O(II) peak gradually increased from 0.07 to 0.111 after 9600 s of illumination, indicating that the UV illumination under vacuum leads to the chemical modification by evolution of the oxygen-vacancy-related defects on the surface of BaSnO₃ (Fig. R6a). As a reverse process, *in situ* characterization using APXPS confirmed that exposure of oxygen-deficient BaSnO₃ films to oxygen atmosphere at 350 °C in darkness caused the area fraction of O(II) peak to decrease and gradually recover to the original states.

In addition, Lee *et al.* confirmed the existence of oxygen-related defects at the surface from cross-sectional scanning transmission electron microscopy (STEM) by comparing high-angle annular dark-field (HAADF) and low-angle annular dark-field (LAADF) signals in vacuum-illuminated BaSnO₃ epitaxial films with nearly perfect arrangement of atoms with cubic symmetry along the [100] zone axis (Figure R6b)^{R31}. The contrast-intensity profiles (yellow rectangles) show distinct change of contrast near the surface of vacuum-illuminated BaSnO₃ epitaxial films between HAADF and LAADF: contrary to almost identical HAADF contrast across the surface, LAADF contrast showed a higher contrast within ≈ 3 nm from the surface than in the bulk area. The difference occurs because the LAADF signal is more sensitive to the strain field from point defects (i.e., V_O) than the HAADF signal. This experiment also verified that the increase in the conductance of the BSO film under UV irradiation is related to the oxygen vacancies generated on the surface.

Our experimental phenomena are fully consistent with the results of Lee *et al.* so it is reasonable to conclude that the conductance change produced by the BSO channel under UV light is caused by oxygen vacancies. When the UV dose [UV dose (mJ/cm²) = UV Intensity (mW/cm²) × Exposure Time (s)] is low, the oxygen vacancy concentration is low, and its impact on the BSO channel can be recovered quickly. On the other hand, when the UV dose is high, the oxygen vacancy concentration increases, resulting in non-volatile conductance changes in the BSO channel.

Fig R7. Synchrotron and STEM analysis of BaSnO₃ thin films during vacuum illumination^{R17}. **a**, In situ APXPS O 1s spectra of BaSnO₃ epitaxial films with ultraviolet-light illumination under vacuum at room temperature (left) and without illumination in oxygen atmosphere at 350 °C (right), with deconvoluted spectra related to lattice oxygen (O(I) at 529.9 eV) (gray shaded area) and defective oxygen (O(II) at 531.3 eV) (red shaded area). **b**, HAADF-STEM (left, top figure) and LAADF-STEM (left, bottom figure) image of vacuum-illuminated BaSnO₃ epitaxial films along the [100] zone axis (scale bar = 5 nm). Contrast-intensity profiles (right) from the yellow rectangles in HAADF-STEM (black line) and LAADF-STEM (red line) images.

5. We appreciate the reviewer to recommend the literature on three-terminal synaptic device. In that report, the authors fabricated ZnO/Ta₂O₅ bottom-gated thin film transistors (TFTs) via Radio Frequency sputtering^{R33}. The device achieved reversible control by modulating oxygen ions inside the channel through gate voltage stimulation. The positive gate bias on the bottom gate electrode attracted the oxygen ions toward the gate/gate insulator interface and repelled the oxygen vacancies toward the channel/gate insulator interface resulting in an oxygen-vacancy-rich or poor region toward the channel interface or gate interface, thereby increasing the device conductance. When a negative voltage was applied, the opposite process occurred.

Electrical modulation through oxygen ions (oxygen vacancies) is common in oxide materials^{R34, R35}, and its good non-volatile properties are also well suitable as an artificial synaptic device^{R36}. In contrast, the non-volatile behavior of our BSO-EGT originated from proton insertion and extraction under electrical stimulation. The basic

physical mechanism is completely different between that study and our present work. It should be noted that our proposed mechanism can readily facilitate the transition between volatile and non-volatile behaviors by using both electric double layer and proton migration phenomena.

Furthermore, our devices integrate multimodal sensing, memory, and processing functions and can emulate switchable short- and long-term plasticity behaviors under optical and electrical stimulation. Thus, this device can simulate the function of human audio-visual integration, demonstrating the potential to mimic the superiority of biological multisensory recognition. Therefore, our BSO-EGT device may be a promising building block for multimodal reconfigurable transistors and provides a feasible solution to deal with complex real-world tasks such as machine vision, autonomous driving, and human-computer interaction.

Corresponding reference

- R23. Bisri SZ, *et al.* Endeavor of Iontronics: From Fundamentals to Applications of Ion-Controlled Electronics. *Advanced Materials* 29, 1607054 (2017).
- R24. Yuan H, *et al.* High-Density Carrier Accumulation in ZnO Field-Effect Transistors Gated by Electric Double Layers of Ionic Liquids. *Advanced Functional Materials* 19, 1046-1053 (2009).
- R25. Prasad B, *et al.* Integrated Circuits Comprising Patterned Functional Liquids. *Advanced Materials* 30, 1802598 (2018).
- R26. Ji H, Wei J, Natelson D. Modulation of the Electrical Properties of VO₂ Nanobeams Using an Ionic Liquid as a Gating Medium. *Nano Letters* 12, 2988-2992 (2012).
- R27. Lu N, *et al.* Electric-field control of tri-state phase transformation with a selective dual-ion switch. *Nature* 546, 124-128 (2017).
- R28. Nishioka D, *et al.* Edge-of-chaos learning achieved by ion-electron-coupled dynamics in an ion-gating reservoir. *Science Advances* 8, eade1156 (2022).
- R29. Wang S, *et al.* An organic electrochemical transistor for multi-modal sensing, memory and processing. *Nature Electronics* 6, 281-291 (2023).
- R30. Liang X, *et al.* Multimode transistors and neural networks based on ion-dynamic capacitance. *Nature Electronics* 5, 859-869 (2022).
- R31. Lee Y, *et al.* Reversible Manipulation of Photoconductivity Caused by Surface Oxygen Vacancies in Perovskite Stannates with Ultraviolet Light. *Advanced Materials* 34, e2107650 (2022).
- R32. De Souza, R.A. Oxygen Diffusion in SrTiO₃ and Related Perovskite Oxides. *Advanced Functional Materials*, 25: 6326-6342 (2015).
- R33. Balakrishna Pillai P, De Souza MM. Nanoionics-Based Three-Terminal Synaptic Device Using Zinc Oxide. *ACS Applied Materials & Interfaces* 9, 1609-1618 (2017).
- R34. Peng HY, *et al.* Effects of electrode material and configuration on the characteristics of planar resistive switching devices. *APL Materials* 1, (2013).

R35. Mou, Xing, *et al.* Analog memristive synapse based on topotactic phase transition for high-performance neuromorphic computing and neural network pruning. *Science Advances* (2021): eabh0648.

R36. Li, Ge, *et al.* Photo-induced non-volatile VO₂ phase transition for neuromorphic ultraviolet sensors. *Nature Communications* (2022): 1729.

Changes made:

To better elaborate the intrinsic mechanisms of BSO-EGT under different stimuli, we added “Previous studies have verified that oxygen vacancies will be generated in BSO films under UV irradiation, resulting in an increase of film conductance. Lee *et al.* used ambient-pressure X-ray photoemission spectroscopy (APXPS) for *in-situ* characterization to monitor the origin of generated defects and proved that the UV illumination under vacuum leads to chemical modification by evolution of the oxygen-vacancy-related defects on the surface of BaSnO₃³⁹. And the result of cross-sectional scanning transmission electron microscopy (STEM) in vacuum-illuminated BaSnO₃ epitaxial films also confirmed the existence of oxygen-related defects at the surface³⁹. When the UV dose [UV dose (mJ/cm²) = UV Intensity (mW/cm²) × Exposure Time (s)] is low, the oxygen vacancy concentration is low, and its impact on the BSO channel can be recovered quickly. On the other hand, when the UV dose is high, the oxygen vacancy concentration increases, resulting in non-volatile conductance changes in the BSO channel.” [Paragraph 3, page 6]

We changed “Supplementary Figure 15a shows that hydrogen ions appear inside the electrically modulated films, and the ion concentration rose significantly as the voltage increased. The addition of a small volume of D₂O to the IL resulted in the presence of D⁺ signal in the electrically-modulated film, which was not observed inside the film without modulation (Supplementary Figure 15b).” to “Supplementary Figure 15a shows that hydrogen ions appear inside the electrically modulated films, and the hydrogen ion concentration rose significantly as the voltage increased. We added a small volume of D₂O to the IL, in which D⁺ just acts as an isotope marker to show the source of the protons (H⁺). The addition of D₂O to the IL resulted in the presence of D⁺ signal in the electrically-modulated film, which was not observed inside the film without modulation (Supplementary Figure 15b).” [Paragraph 3, page 8]

We changed “For a positive V_G lower than the threshold value of the hydrolysis reaction, anions and cations in ILs accumulate at the IL/gate and IL/channel interfaces, respectively. Thus, the introduction of electrons through EDL can increase channel conductance effectively.” to “The volatile response of BSO-EGT at a lower V_G originated from the rapid movement of anions and cations in ILs under an electric field, which produced a strong accumulation of space charge at the interface between the electrolyte and the channel, called a Helmholtz layer or EDL⁴⁸. Accumulation occurs due to blocking ions in the solid channel. In order to balance the EDL formed at the interface, an accumulation of electrons occurs inside the channel. Therefore, the conductance of the channel will change. When the external bias voltage is removed, the

ions at the interface spontaneously migrate back into the ionic liquid. The electrical double layer disappears, so the channel conductance returns to its original state.” [Paragraph 2, page 7]

We added “The conductance of the BSO channel showed non-volatile changes under high voltage stimulation, and there is a peak in the gate current of the transfer characteristic curve at approximately 1.3 V (Supplementary Figure 10a), which is related to the hydrolysis reaction⁴⁹. This is because, in addition to the presence of EDL at the interface, H⁺ originating from the trace water containing in the ionic liquids would be injected into the film when the positive voltage exceeded the critical voltage. Protons (H⁺) produced by hydrolysis can be driven to the channel interior, resulting in strong interactions with the solid material^{48, 50}. Therefore, the non-volatility of the device came from the migration of protons generated by hydrolysis in the ionic liquid into the oxide film under the positive gating, causing a non-volatile increase in the channel conductance.” [Paragraph 2, page 8]

We changed “In order to analyze the origin of the non-volatility, secondary ion mass spectrometry (SIMS) was performed on the BSO films after applying different electrical stimuli.” to “To verify this mechanism, secondary ion mass spectrometry (SIMS) was performed on the BSO films after applying different electrical stimuli.” [Paragraph 3, page 8]

Corresponding references were added in the revised manuscript.

48. Bisri SZ, *et al.* Endeavor of Iontronics: From Fundamentals to Applications of Ion-Controlled Electronics. *Advanced Materials* 29, 1607054 (2017).

49. Lu N, *et al.* Electric-field control of tri-state phase transformation with a selective dual-ion switch. *Nature* 546, 124-128 (2017).

50. Yuan HT, *et al.* Hydrogenation-Induced Surface Polarity Recognition and Proton Memory Behavior at Protic-Ionic-Liquid/Oxide Electric-Double-Layer Interfaces. *Journal of the American Chemical Society* 132, 6672-6678 (2010).

Point 5.

Their dataset description is very confusing. It needs to be clarified which data was used in training and which was used in testing. How many samples were there in each category? Is there a citation for the database? What precautions were taken to avoid over-fitting?

Response:

Thanks for the reviewer’s valuable suggestions. Following your suggestions, we will elaborate on the audio dataset in more detail.

The audio dataset is generated based on the built video sub-dataset. The video classification task uses the public EgoGesture dataset^{R37, R38}, which contains a total of 83 different gesture. Each category contains a large number of samples and each sample

in this dataset contains dozens of chronologically arranged frames describing the corresponding gesture.

We selected 5 gestures (“scroll hand toward left”, “rotate fists clockwise”, “zoom in with fingers”, “sweep circle” and “move fist downward”) from 83 categories (Supplementary Figure 21) and selected 250 samples from each corresponding category to construct a sub-dataset with a sample size of 1250.

Through 25 different public AI sound sources, the names of the five selected actions from EgoGesture dataset are converted into speeches, and 125 original samples are generated. After pre-processing and adding noise, we were able to expand the sample size to 1250.

The ratio of training set and test set of video and audio dataset both are 1:1 which means each category contains 625 samples. We cited relevant references for the EgoGesre dataset in the manuscript^{R37, R38}. Since the audio database was built by ourselves, there are no references related to the database in the manuscript.

During the training process of the neural network, we limit the number of iterations. Within this range, no obvious drop in accuracy was observed, so we believe that overfitting is avoided.

Corresponding references:

R37. Y. Zhang, *et al.* EgoGesture: A New Dataset and Benchmark for Egocentric Hand Gesture Recognition, *IEEE Transactions on Multimedia*, vol. 20, no. 5, pp. 1038-1050, May 2018.

R38. Congqi Cao, Yifan Zhang, Yi Wu, Hanqing Lu, Jian Cheng; *Proceedings of the IEEE International Conference on Computer Vision (ICCV)*, 2017, pp. 3763-3771.

Changes made:

We changed the description of audio dataset used in the multimodal dynamic gesture recognition task and added more details in the revised manuscript:

We changed “With the help of a text-to-speech conversion program, we collected 5 types of actions spoken by 25 people as speech samples.” to “Through 25 different public AI sound sources and text-to-speech conversion program, the names of the five selected actions are converted into speech, and 125 original samples are generated.” [paragraph 2, page 13]

We changed “On this basis, 10% random noise is added to the audio input signal to simulate possible bit error ratio (BER) during digital signal transmission, and the number of samples was also expanded to 1250.” to “On this basis, 10% random noise is added to the audio input signals to simulate possible bit error ratio (BER) during digital signal transmission, and the number of samples in audio dataset was also

expanded to 1250 to match the size of corresponding video dataset. More details about dataset construction can be found in the Methods (Dataset construction).” [paragraph 2, page 13]

We also significantly modified the content of the dataset construction section. The original and modified version are as follows:

Original version:

Five gestures were selected from eighty-three categories of EgoGesture dataset, and the first 250 samples of each category were retained to construct a sub-dataset of size 1250, with a 1:1 ratio of training and test sets samples. The frames were extracted from each video sample in the EgoGesture sub-dataset to obtain dozens of images. In order to compress the data volume, four frames were randomly selected from each video sample and arranged in order to form the initial inputs.

modified version:

The video classification task uses the public EgoGesture dataset, which contains a total of 83 different gesture. Each category contains a large number of samples and each sample in this dataset contains dozens of chronologically arranged frames describing the corresponding gesture. We selected 5 gestures from 83 categories and the first 250 samples from each corresponding category are retained to construct a sub-dataset with a sample size of 1250.

The audio dataset is generated based on the built video sub-dataset. Through 25 different public AI sound sources, the names of the five selected gestures from EgoGesture dataset are converted into speeches, and 125 original samples are generated. After pre-processing and adding noise, the volume of the audio dataset also expands to 1250. In the video and audio datasets, the ratio of training set and test set is 1:1 which means each category contains 625 samples. [Method, Dataset construction]

Point 6.

The video classification section similarly can be improved in terms of readability. Which parts of the ANN were implemented in software and which in hardware is not easy to distinguish.

Response:

We are grateful to the reviewer for the valuable questions on our manuscript. Based on your question, we will give a more detailed discussion on the video classification task.

We conducted 4-bit binary streams volatility testing of BSO-EGT in different modes (“LLLL”, “LLLE”, “LLEE”, “LEEE”, “EEEE”; Figure 3 and Supplementary Figure 16-17). Based on the extracted experimental results, the reservoir model of the device under different types of stimulation was constructed. In addition, we also tested the non-

volatile characteristics of BSO-EGT, namely the long-term potentiation/depression (LTP/LTD) curves (Figure 2h), from which the conductance changes were modeled as the synaptic weight update method in the neural network. After that, we demonstrated the video classification task through Matlab based on the built models. Software simulation based on building a suitable physical model is a very common and effective way in the field of neuromorphic computing^{R16-R18, R39}.

Each sample in the EgoGesture dataset contains dozens of frames. To compress the amount of data, we randomly extracted four frames in chronological order. The size of each frame is 320 pixels \times 240 pixels, which can be represented by a binary matrix with the size of [320, 240] after binarization. After expanding the matrix, we can get a one-dimensional column vector with the size of [76800, 1]. After merging the vectors corresponding to the four pictures, each final sample input needs to be represented by a matrix of size [76800,4].

Each row of the matrix was encoded with light pulses and fed into the corresponding reservoir. After being processed and compressed by the reservoirs, the input of the ANN is a vector of length 76800. Therefore, a two-layer fully connected artificial neural network with a size of [76800, 5000, 5] was used to perform the video classification recognition task. The ReLu function and the cross-entropy loss function were selected as the activation function of hidden layer and output layer respectively, and the weights were updated based on the backpropagation algorithm. The audio classification used the same approach, while the difference is that the size of the network. This task only requires a single-layer neural network, and the activation function of the output layer is also a cross-entropy loss function.

Corresponding references:

R39. Jiang, T, *et al.* Tetrachromatic vision-inspired neuromorphic sensors with ultraweak ultraviolet detection. *Nature Communications* 14, 2281 (2023).

Changes made:

To describe the implementation of video classification task, we changed “The pixel matrix of each frame was reorganized into a single column vector array after binarization. Then, the four frames of a sample were combined into a new matrix of size [76800, 4], and each row of the matrix was encoded using light pulses and fed into the corresponding reservoir.” to “Each sample in the EgoGesture dataset contains dozens of frames. To compress the amount of data, we randomly extracted four frames in chronological order. The size of each frame is 320 pixels \times 240 pixels, which can be represented by a binary matrix of size [320, 240] after binarization. Then, we expanded the matrix to get a one-dimensional column vector with the size of [76800, 1]. After merging the vectors corresponding to the four frames, every final input needs to be represented by a matrix of size [76800,4]. Then, each row of the matrix was encoded with light pulses and fed into the corresponding reservoir.” [Method, Video classification]

We changed “The ReLu function was selected as the activation function of the network, and the weights were updated based on the backpropagation algorithm.” to “The ReLu function and the cross-entropy loss function were selected as the activation functions of the hidden layer and readout layer respectively, and the weights were updated based on the back propagation algorithm.” [Method, Video classification]

We added “This task only requires a single-layer neural network, and the activation function of the output layer is also a cross-entropy loss function.” [Method, Video classification]

Point 7.

What is the meaning of pollution degree and tackle? Please define

Response:

We are very thankful to the reviewer for the valuable suggestion. Since our expression about pollution is unclear, which makes it difficult to understand this part in the manuscript, we will define pollution degree as follows:

Uncontaminated pictures content can be acquired through optical detection (vision), while pictures contaminated by pigments are no longer visible, but can be perceived through touch^{R13}. The pressure information can be converted into electrical signals for processing through pressure sensors. Any picture in the Fashion-MNIST data set contains 784 (28×28) pixels, so the “pollution degree” or “invisible degree” indicates how many proportions of the pixels in the picture are contaminated. The image information of the polluted part can no longer be obtained through optical perception, but can only be obtained through electrical signals.

Changes made:

In order to better understand Figure 4 and pollution degree, we added “The “pollution degree” or “invisible degree” indicates how many proportions of the pixels in the picture are contaminated. For ease of demonstration, we omit the sensing and processing steps of the piezoelectric sensors. The image information of the polluted part can no longer be obtained through optical perception, but can only be obtained through electrical signals.” [paragraph 2, page 11]

Point 8.

Very difficult to understand what is an audiovisual fusion task. How was this implemented? How was the input converted into voltage and light signals? The authors claim the improvement in combined efficiency for audio + video is due to multimode information processing. I am not convinced of that. Most articles report high efficiency for audio tasks due to limitations of the size of their dataset used in training, whereas for video the efficiencies reported are upto 94%. I fear the authors citations are incomplete on this front.

Response:

Thanks to the reviewers for their valuable comments. Based on your questions, we have added supplementary discussion to audio-visual fusion recognition:

Audio-visual integration is to use visual and auditory information to achieve a more comprehensive information acquisition process. Mixed recognition of two types of information showed higher recognition accuracy than single information modalities. Inspired by organisms using multiple sensory information to achieve high-level cognitive processes^{R40-R43}, more and more researches aim to simulate this process on hardware. The mechanisms of multisensory integration have therefore been studied. The article published in *Nature Electronics*^{R12} compared the recognition accuracy of different types of fusion mechanisms. In this work, we use a decision-level fusion approach to achieve more comprehensive information acquisition through visual and auditory information.

As mentioned in Point 6, we established device parameter models based on the actual experiment results of the BSO-EGT devices. Based on these models, we constructed an optical reservoir array, an electrical reservoir array, and an electrical synapse array respectively to implement optical sensing & pre-processing, electrical sensing & pre-processing, and electrical computing & storage functions. Therefore, we can realize the integration of sensing, memory and processing based on the same device.

Similar to the contaminated image processing process, the samples used for the video classification task are multi-frame pictures with temporal information. It can be understood as mapping the pixel matrix into a light pulse matrix, which is illuminated on the optical reservoir array to achieve information perception; audio samples can be understood as converting sound wave signals into electrical signals through recording equipment, and then applying electrical signals to the electric reservoir array for pre-processing. After pre-processing in the reservoirs, all results need to be sent to the subsequent artificial neural network for subsequent post-processing, that is, using electrical synapse arrays for storage and calculation.

Optical reservoirs and electrical synaptic arrays are used when performing video classification tasks. While classifying audio samples, electrical reservoirs and synapses are used for calculations. When performing audio-visual fusion recognition, three arrays are used for processing.

Current research focusing on video/audio classification uses complex feature extraction technology and deep neural networks (multi-layer convolutional networks, attention mechanisms, multi-head mechanisms, etc.), which can achieve very high recognition accuracy^{R44-R46}. Our work focuses on physical modeling based on BSO-EGT devices and analyzing performance in multi-mode recognition tasks. Therefore, we did not use special feature extraction technologies, but only utilized the inherent timing processing

capabilities of the physical reservoirs, and a simple double-layer (single-layer) neural network (Method, Video classification) for video (audio) classification. If more complex deep learning algorithms are used, the overall recognition rate of audio-visual mixed recognition tasks can be improved to a higher level. Since the length of the input vector is very long (5000 for audio and 76800 for video) and the neural network architecture is very simple, the accuracy of video or audio classification alone will be relatively low. However, if the inputs of two different modes are processed through their respective reservoirs and neural networks, and then integrated and analyzed through the fusion-layer neural network, the final recognition accuracy is greatly improved compared to the other two separate recognition results, indicating the superiority of audio-visual fusion recognition. At the same time, the result also demonstrated the good potential of BSO-EGT with multi-modal sensing and good reconfigurability in reservoir computing and neuromorphic computing.

Corresponding references:

- R40. Li G, *et al.* Skin-inspired quadruple tactile sensors integrated on a robot hand enable object recognition. *Sci. Robot.* 5, eabc8134 (2020).
- R41. Wang M, *et al.* Gesture recognition using a bioinspired learning architecture that integrates visual data with somatosensory data from stretchable sensors. *Nature Electronics* 3, 563-570 (2020).
- R42. Jiang C, *et al.* Mammalian-brain-inspired neuromorphic motion-cognition nerve achieves cross-modal perceptual enhancement. *Nature Communications* 14, 1344 (2023).
- R43. Liu M, *et al.* A star-nose-like tactile-olfactory bionic sensing array for robust object recognition in non-visual environments. *Nature Communications* 13, 79 (2022).
- R44. Delbrouck, J. B, *et al.* A transformer-based joint-encoding for emotion recognition and sentiment analysis. arXiv preprint arXiv:2006.15955 (2020).
- R45. Hu G, *et al.* Unimse: Towards unified multimodal sentiment analysis and emotion recognition. arXiv preprint arXiv:2211.11256 (2022).
- R46. Pei M. *et al.* Power-Efficient Multisensory Reservoir Computing Based on Zr-Doped HfO₂ Memcapacitive Synapse Arrays. *Advanced Materials*, 35(41), 2305609.

Changes made:

In order to facilitate the understanding of the implementation of audio-visual fusion tasks, we added “Similar to the static image processing process, the multi-frame pictures with temporal information can be mapped to the pixel matrix into a light pulse matrix, which is illuminated on the optical reservoir array to achieve information perception. More details can be found in Method.” [paragraph 3, page 12]

We changed “In such network, separate models are trained based on audio and video datasets, and the classification results of the pre-trained model are fed into the post-decision network for final inference.” to “The audio and video datasets are processed

through their respective reservoirs and neural networks, and then integrated and analyzed through the fusion-layer neural network.” [Paragraph 3, page 13]

Point 9.

In summary, the methods of this article are not completely described. There needs to be more depth to their interpretation of the device mechanism and learning efficiency. There needs to be more substantive comparison with other state of the art implementations and context. What is the motivation for this type of a complex device with relatively less impressive performance?

Response:

We are grateful to the reviewer’s comments. In the revised manuscript, we prepared new BSO samples under different growth conditions and tested new devices. After efforts, we now can address all the reviewer’s concerns and update the corresponding experimental data in the revised manuscript. For a more detailed description of the device mechanism, we have made corresponding changes in the revised manuscript and supplemented more experiments to strengthen our conclusion.

Next, we would like to discuss more about the motivation of our work. The biological brain integrates multimodal perception, memory and processing functions and excels in decision-making on both dynamic and static information. Bio-inspired artificial neuromorphic systems are expected to solve the bottleneck faced in the traditional computer architecture, that is high latency and large energy consumption caused by data shuffling between memory and processing units, and have therefore received considerable attention.

Reconfigurable neuromorphic transistors, which can be employed to emulate different types of biological analogues in a single device^{R18}, are important for manufacturing compact and efficient neuromorphic computing architecture, but its design and implementation remain challenging due to the need for opposing physical mechanisms to achieve different functions. For example, the hardware implementation of physical reservoirs, that mimic human receptors for pre-processing, should have volatility; but artificial synapses, that emulate the cerebral cortex for post-processing, typically require good non-volatility. At present, it is still very difficult to integrate both functions in a single device. In previous studies, researchers generally used ideal software simulation models to replace artificial synaptic devices to perform weight storage and update functions in artificial neural networks^{R13}, or used sensors to convert different types of information into a single type of electrical signal for follow-up processing^{R19, R42}. The article published in *Nature Electronics* uses cameras and pressure sensors to achieve multi-modal perception of vision and gesture movements^{R12}. Research published in *Nature Communications* used different types of sensors to simulate the process of biological recognition using touch and smell^{R11}. And information processing

requires additional edge computing devices in most researches. This essentially limits the potential for future applications of neuromorphic hardware systems.

In this work, we proposed a novel optoelectronic transistor that can switch between short-term and long-term memory under optical and electrical stimuli. The experimental results show that our device can be reconfigured to meet the requirements of both volatile and non-volatile behavior. Therefore, the proposed transistor is capable of realizing reconfigurable functions between physical reservoir and artificial synapse in a single device.

Our work enables multimodal sensing, storage and processing in a single device. It is no longer necessary to use additional sensing units for information conversion or additional cores for task processing. This demonstrates that multiple types of information processing are possible through homogenous integration of devices. Furthermore, our scheme provides a feasible solution to deal with complex real-world tasks such as machine vision, autonomous driving, and human-computer interaction.

Changes made:

To highlight the novelty of our manuscript, we added “Previous studies usually used ideal software simulation models to replace artificial synaptic devices to perform weight storage and update functions in artificial neural networks³⁰; or used sensors to convert different types of information into a single type of electrical signal for follow-up processing^{29, 31}. And information processing requires additional edge computing devices in most researches. This essentially limits the potential for future applications of neuromorphic hardware systems.” [Paragraph 1, page 4]

We added “Compared with previous studies, our BSO-EGT has the advantages of precise regulation and comprehensive functions in terms of reconfigurability (Supplementary Table R1). At the application demonstration level, we verified its multi-sensory integration capabilities and realized advanced neuromorphic applications with the multimode sensing, storage and processing capabilities of a single device.” [Paragraph 1, page 14]

We also added a new table to compare with recent works about neuromorphic transistors and multimodal systems.

Ref.	On/off ratio	Optical reconfigurability	Electrical reconfigurability	Reservoir+Synapse	Multisensory integration	Task type
1	$> 10^3$	/	Yes	Yes	/	static
2	$> 10^6$	Potentially Yes	Yes	/	Tactile + Visual	static
3	$> 10^6$	/	Yes	Yes	/	Dynamic
4	$> 10^3$	/	/	/	Ocular+Vestibular	Dynamic
This work	$> 10^6$	Yes	Yes	Yes	Audio + Visual	Dynamic

Supplementary Table R1. Comparison with state-of-the-art multimodal neuromorphic transistors.

Corresponding references

1. John RA, *et al.* Reconfigurable halide perovskite nanocrystal memristors for neuromorphic computing. *Nature Communications* 13, 2074 (2022).
2. Liu K, *et al.* An optoelectronic synapse based on α -In₂Se₃ with controllable temporal dynamics for multimode and multiscale reservoir computing. *Nature Electronics* 5, 761-773 (2022).
3. Wang S, *et al.* An organic electrochemical transistor for multi-modal sensing, memory and processing. *Nature Electronics* 6, 281-291 (2023).
4. Jiang C, *et al.* Mammalian-brain-inspired neuromorphic motion-cognition nerve achieves cross-modal perceptual enhancement. *Nature Communications* 14, 1344 (2023).

Reviewer #3 (Remarks to the Author):

Authors suggested an EGT-based synaptic transistor, utilizing BaSnO₃ films in an attempt to enhance the photoresponse characteristics of neuromorphic devices. However, there is some issue and lack of novelty to be published in “Nature communication” Therefore, my recommendation for this manuscript is “transfer to another journal in nature publication”. The followings are my comments in detail for this manuscript.

Response:

Thank you very much for your hard-work in reviewing our manuscript. The novelty we would like to highlight here is the realization of a device that is precisely reconfigurable between volatile reservoirs and non-volatile synapses, thus homogeneously integrating multimodal sensing, storage and processing functions. Thanks to the reviewers for recommending three literatures. After careful reading and analysis, we can draw a conclusion that these three articles are essentially different from our work. All three articles focus on achieving non-volatile synaptic functions, whereas our work achieves precisely reconfigurable functions between volatile reservoirs and non-volatile synapses (Table R1). More discussion about Table R1 is given in the response to the reviewer’s comments. Moreover, the works recommended by the reviewers mainly implement recognition of static images, while our work focuses on the processing of dynamic information.

In summary, by comparing the information diversity, functional complexity, and task difficulty of the device, it can be found that our work is significantly different with the articles recommended by the reviewer. The point-by-point responses to the reviewer’s comments are given in the following.

	Optical stimulation		Electrical stimulation			Function	Task type
	Volatile	Non-volatile	Volatile	Non-volatile	Switching method		
Ref. [1] Adv. Mater. 2023	√	√	√	√	Pulse number	Synapse	Static
Ref. [2] Appl. Phys. Lett. 2023	×	√	√	×		Synapse	Static
Ref. [3] Adv. Mater. 2022	√	√	×	×		Synapse	Static
Our work	√	√	√	√	Voltage amplitude (precise)	Reservoir + Synapse	Dynamic + Static

Table R1. Compare with articles recommended by the reviewer in terms of device response, reconfigurability, functionality, and task types.

Point 1.

Firstly, the EGT-based synaptic transistor presented in the study does not significantly differentiate itself from existing works in the field, particularly concerning the use of BaSnO₃ films. The lack of demonstrable superiority in aspects such as photoresponse and applicable light wavelengths underscores a notable absence of innovation.[1]

Response

Thanks for the reviewer's comments on our manuscript.

1. Most existing works exploit only the non-volatile properties of the device to achieve synaptic functionality^{R47-R49}. Some others use the volatility of the device to implement the reservoir function with an ideal model to update the weight of the readout layer^{R50-R52}. The reservoir can realize high-dimensional mapping of data, compress the amount of data, and speed up task processing, and has obvious advantages in the task of processing sequence information^{R53-R55}. Homogeneously integrating reservoir and synaptic functions is difficult but urgently needed in terms of integration and reduced manufacturing complexity^{R56, R57}. Reservoir requires devices to exhibit volatile properties, while synaptic function requires devices to exhibit nonvolatile properties. Therefore, integrating these two opposing dynamics processes is very challenging. By changing the amplitude of the voltage bias and UV irradiation, our proposed device can be precisely reconfigurable between volatile and nonvolatile modes, combining both reservoir and synaptic functions. Therefore, our work significantly differentiates from most of existing works in this field.

2. We agree with the reviewer that the choice of material is very important for future application. In this work, we mainly propose a design principle for a kind of reconfigurable device. Channel materials include but are not limited to BaSnO₃. A variety of oxide materials with responses to light can serve as channels. For example, InGaZnO₄ (IGZO), which is widely concerned by industry, can also be an option.

IGZO has been mainly applied in industry due to its optical transparency, low processing temperature, and compatibility with various gate insulators^{R58, R59}. It occurs the band-to-band excitation, the oxygen vacancies ionization, and the metastable peroxides formation in the a-IGZO semiconductor during the light illumination illumination^{R60}. Due to the high density of the trap states in the gap, the high-quality a-IGZO films have a small conduction band tail (~2.3 eV)^{R61}, resulting in broad spectral responses. The use of amorphous IGZO in thin film transistors offers numerous advantages, including excellent uniformity, high mobility, high switching current ratio, and large-scale processing^{R62}.

We grew 15 nm a-IGZO film on SiO₂/Si substrates at room temperature under O₂ pressure of 5 Pa through pulsed laser deposition. Afterwards, the film was annealed at 300°C for 1 hour. X-ray photoelectron spectroscopy (XPS) results showed clear

characteristic peaks of In 3d, Ga 2p and Zn 2p, respectively, indicating that the IGZO film is of good quality (Figure R8).

Figure R8. X-ray photoelectron spectroscopy (XPS) spectra of **a**, In 3d, **b**, Ga 2p and **c**, Zn 2p in an IGZO thin-film.

We used IGZO as the channel material to fabricate an electrolyte-gated transistor (IGZO-EGT) with the same device structure. Based on the same experiment scheme, IGZO showed the potential to achieve the same functionalities we demonstrated in the manuscript with BSO film. IGZO-EGT also demonstrates the ability to accurately switch between volatile and non-volatile modes affected by voltage amplitude (Figure R9). The response of IGZO-EGT at green (532nm), blue (450nm) and ultraviolet (375nm) light wavelengths were tested (Figure R10). The experimental results showed IGZO-EGT had wide applicable light wavelengths, and can address the reviewer's concerns about the narrow response range of BSO-EGT. Moreover, a clear non-linear relaxation process was observed in IGZO-EGT after the electrical or optical stimulation was removed, indicating its potential to realize reservoirs. Therefore, using the design concept we proposed in this work, electrolyte transistors using IGZO as the channel material can also achieve the expected reservoir and artificial synapse functions, and then complete complex real-world tasks such as audio-visual fusion recognition.

Figure R9. **a**, Schematic illustration of the neuromorphic transistor that can be stimulated using optical and electrical signals. The InGaZnO₄ film serves as a channel between the source (S) and drain (D) electrodes, and IL is used as the gating medium. **b**, Transfer curves of the IGZO transistor measured with $V_{SD} = 0.5\text{ V}$. **c**, The channel current controlled through a series of V_G pulse with a pulse width of 1s and different amplitudes which shows the transition of the PSC from volatile to non-volatile property. **d**, Channel current under voltage stimulation of +1 V and +3 V. **e**, +1 V voltage pulse stimulation with different pulse durations. **f**, +2.5 V voltage pulse stimulation with different pulse durations. **g**, Pulse-switching characteristics. The high conductance state is generated by $V_G = +2.5\text{ V}$, and the low conductance state is generated by $V_G = -2.5\text{ V}$. **h**, The non-volatile multi-level conductance retention properties. The multi-states are produced by a series of V_G pulses with different pulse numbers ($V_G = +2.5\text{ V}$). **i**, Cyclic controlled LTP using V_G (equally spaced from +1.5 to +3.5 V, duration of 1 s, spaced 1 s apart) and LTD using V_G (equally spaced from -1.5 to -3.5 V, duration of 1 s, spaced 1 s apart) for 10 pulses.

Figure R10. Photoresponse characteristics of IGZO-EGT. **a**, Evolution of channel current under UV light (375 nm) irradiation of different durations (7 mW/cm²). **b**, Evolution of channel current under blue light (450 nm) irradiation of different durations (70 mW/cm²). **c**, Evolution of channel current under green light (532 nm) irradiation of different durations (70 mW/cm²). **d-f**, The evolution of channel current stimulated by different numbers of light pulses, under UV (375nm), blue(450nm) and green light (532nm) illumination

3. On the other hand, the reviewer's questions prompted us to further optimize the performance of BaSnO₃ devices.

Most neuromorphic sensors operate in the visible light range, presenting the possibility of replacing the human visual system. But the ultraviolet band also plays an irreplaceable role in human life, such as in health care, rocket warning and missile detection. Therefore, it is important to study detectors that work in the ultraviolet band and are not affected by illumination in other optical bands. BSO has a wide optical bandgap (~ 3eV), thus showing good potential as an ultraviolet photodetector.

The best room temperature values of the mobility and conductivity in the model perovskite oxide semiconductor SrTiO₃ have remained below 10 cm² V⁻¹ s⁻¹ and 500 S cm⁻¹, respectively for over 50 years^{R63}. Recently, Kim *et al.* has shown that BaSnO₃ has higher mobility of 320 cm² V⁻¹ s⁻¹ and a conductivity value of 4×10³ S cm⁻¹ at a carrier concentration of 8×10¹⁹ cm⁻³. BaSnO₃ has therefore attracted significant interest as a possible alternative to indium compounds for transparent electrodes^{R64, R65}.

Considering the photoresponse mentioned by the reviewer, the performance of the device has been improved (Figure R11) by optimizing the growth conditions of BSO. We found that increasing the oxygen pressure during the film growth will lead to a decrease in the film conductivity. This process is attributed to the decrease in the oxygen

vacancy content^{R66}. This can be attributed to the mechanism that ionized oxygen vacancies could provide the additional mobile carriers for conduction, leading to the increased carrier density, and oxygen vacancies can neutralize the negative charges at threading dislocations, which suppresses the contribution of dislocation scattering^{R21}. Films grown under high oxygen pressure conditions can significantly improve their photoresponse because they can generate a larger proportion of oxygen vacancies under ultraviolet light. Under illumination, the light/dark current ratio increased to more than an order of magnitude. All experimental data in the manuscript have been completely updated using devices with more pronounced photoresponse.

Furthermore, the off-state current of newly manufactured device is 0.1 nA, which can reduce power consumption. When applying a 20 ms pulse voltage stimulus (1 V), the power consumption of a single spike is 12 pJ. While the article recommended by the reviewer operates under the condition of $V_{SD} = -10$ V. It is obvious that our device has lower power consumption during the actual test process. From the comparison of electrical synapse performance, the article recommended by the reviewer has a switching ratio of 10^4 in the scanning range of $V_G = \pm 30$ V, but our device has a switching ratio of more than 10^6 in the scanning range of $V_G = \pm 2.5$ V. A conclusion can be drawn that our optimized device has better performance by comparing these data.

Fig R11. EGT with reconfigurable characteristics for multimode sensing. a, Schematic illustration of the neuromorphic transistor that can be stimulated using optical and electrical signals. The BaSnO₃ film serves as a channel between the source

(S) and drain (D) electrodes, and IL is used as the gating medium. **b**, Short-term potential effect stimulated using optical pulses (1s) with a 0.5 V bias to monitor the channel current. **c**, Transition from STM to LTM under different light durations. **d**, Memory decay process after light pulse (70 mW/cm² for 1 s) stimulation, fitted using a double-exponential function. The fitting result shows the coexistence of two physical processes. **e**, Excitatory post synaptic current (PSC) induced via a train of optical pulse (70 mW/cm², 1s with intervals of 1, 5, 20 and 30s) with a 0.5 V bias to monitor the channel current. **f**, The channel current controlled through a series of V_G pulse with a pulse width of 1s and different amplitudes which shows the transition of the PSC from volatile to non-volatile property. **g**, Spike number-dependent plasticity under electrical stimuli (+2 V, 1s) with the same intervals of 1s. **h**, Cyclic controlled LTP using V_G (equally spaced from +1.5 to +3.5 V, duration of 1s, spaced 1s apart) and LTD using V_G (equally spaced from -1.5 to -3.5 V, duration of 1s, spaced 1s apart) for 32 pulses.

Corresponding reference:

R47. Jeong B, Gkoupidenis P, Asadi K. Solution-Processed Perovskite Field-Effect Transistor Artificial Synapses. *Advanced Materials*. 33, e2104034 (2021).

R47. Huang J, *et al.* Flexible, Transparent, and Wafer-Scale Artificial Synapse Array Based on TiO_x/Ti₃C₂T_x Film for Neuromorphic Computing. *Advanced Materials*. 35, 2303737 (2023).

R49. Kumar D, *et al.* Flexible Solution-Processable Black-Phosphorus-Based Optoelectronic Memristive Synapses for Neuromorphic Computing and Artificial Visual Perception Applications. *Advanced Materials*. 35, e2300446 (2023).

R50. Zhang Z, *et al.* In-sensor reservoir computing system for latent fingerprint recognition with deep ultraviolet photo-synapses and memristor array. *Nature Communications*. 13, 6590 (2022).

R51. Tan H, van Dijken S. Dynamic machine vision with retinomorphic photomemristor-reservoir computing. *Nature Communications*. 14, 2169 (2023).

R52. Liu K, *et al.* An optoelectronic synapse based on α -In₂Se₃ with controllable temporal dynamics for multimode and multiscale reservoir computing. *Nature Electronics*. 5, 761-773 (2022).

R53. Moon J. *et al.* Temporal data classification and forecasting using a memristor-based reservoir computing system. *Nature Electronics*. 2, 480-487 (2019).

R54. Milano G. *et al.* In material reservoir computing with a fully memristive architecture based on self-organizing nanowire networks. *Nature Materials* 21, 195-202 (2022).

R55. Zhong Y. *et al.* Dynamic memristor-based reservoir computing for high-efficiency temporal signal processing. *Nature Communications* 12, 408 (2021).

R56. Wang S, *et al.* An organic electrochemical transistor for multi-modal sensing, memory and processing. *Nature Electronics* 6, 281-291 (2023).

R57. John RA, *et al.* Reconfigurable halide perovskite nanocrystal memristors for neuromorphic computing. *Nature Communications* 13, 2074 (2022).

R58. Hays DC, *et al.* Energy band offsets of dielectrics on InGaZnO₄. *Applied Physics Reviews* 4, (2017).

- R59. Jang Y, *et al.* Amorphous InGaZnO (a-IGZO) Synaptic Transistor for Neuromorphic Computing. *Acs Applied Electronic Materials* 4, 1427-1448 (2022).
- R60. Ke S, *et al.* Indium-Gallium-Zinc-Oxide Based Photoelectric Neuromorphic Transistors for Modulable Photoexcited Corneal Nociceptor Emulation. *Advanced Electronic Materials* 7, 2100487 (2021).
- R61. Keisuke Ide, *et al.* Effects of excess oxygen on operation characteristics of amorphous In-Ga-Zn-O thin-film transistors. *Appl Phys Lett* 99, (2011).
- R62. Liu H, *et al.* High Performance and Hysteresis-Free a-IGZO Thin Film Transistors Based on Spin-Coated Hafnium Oxide Gate Dielectrics. *IEEE Electron Device Letters* 44, 1508-1511 (2023).
- R63. Schlom DG, Pfeiffer LN. Oxide electronics: Upward mobility rocks! *Nature Materials* 9, 881-883 (2010).
- R64. Prakash A, *et al.* Wide bandgap BaSnO₃ films with room temperature conductivity exceeding 10⁴ S cm⁻¹. *Nature Communications* 8, 15167 (2017).
- R65. Luo BC, Hu JB. Unraveling the Oxygen Effect on the Properties of Sputtered BaSnO₃ Thin Films. *Acs Applied Electronic Materials* 1, 51-57 (2019).
- R66. Lee Y, *et al.* Reversible Manipulation of Photoconductivity Caused by Surface Oxygen Vacancies in Perovskite Stannates with Ultraviolet Light. *Advanced Materials* 34, e2107650 (2022).

Changes made:

In order to address the concern raised by the reviewer about applicable light wavelength, we added the following section:

Demonstration of wide application wavelength based on IGZO

In this work, we mainly propose a design principle for a kind of reconfigurable device. Channel materials include but are not limited to BaSnO₃. A variety of oxide materials with responses to light can serve as channels. For example, InGaZnO₃ (IGZO), which is widely concerned by industry, can also be an option. IGZO has been mainly applied in industry due to its optical transparency, low processing temperature, and compatibility with various gate insulators^{58, 59}. It occurs the band-to-band excitation, the oxygen vacancies ionization, and the metastable peroxides formation in the a-IGZO semiconductor during the light illumination⁶⁰. Due to the high density of the trap states in the gap, the high-quality a-IGZO films have a small conduction band tail (~2.3 eV)⁶¹, resulting in broad spectral responses. The use of amorphous IGZO in thin film transistors offers numerous advantages, including excellent uniformity, high mobility, high switching current ratio, and large-scale processing⁶².

We grew 15 nm a-IGZO film on SiO₂/Si substrates and X-ray photoelectron spectroscopy (XPS) results showed clear characteristic peaks of In 3d, Ga 2p and Zn 2p, respectively, indicating that the IGZO film is of high quality (Supplementary Figure 24). IGZO was used as the channel material to fabricate an electrolyte-gated transistor (IGZO-EGT) with the same structure. Based on the same experiment scheme, IGZO showed the potential to achieve the same functionalities we demonstrated with BSO

film. IGZO-EGT also demonstrates the ability to accurately switch between volatile and non-volatile modes affected by voltage amplitude (Supplementary Figure 25). The response of IGZO-EGT at green (532nm), blue (450nm) and ultraviolet (375nm) light wavelengths was also tested (Supplementary Figure 26) and showed the wide applicable light wavelengths of IGZO-EGT. Moreover, a clear nonlinear relaxation process was observed in IGZO-EGT after the electrical or optical stimulation was removed, indicating its potential to realize reservoirs. Therefore, using the design concept we proposed, electrolyte transistors using IGZO as the channel material can also achieve reconfigurable reservoir and artificial synapse functions, and then complete complex tasks such as audio-visual fusion recognition. [Paragraph 2-3, page 14]

We also added three supplementary figures (Supplementary Figure 24-26).

Supplementary Figure 24. X-ray photoelectron spectroscopy (XPS) spectra of a, In 3d, b, Ga 2p and c, Zn 2p in an IGZO thin-film.

Supplementary Figure 25. Electrical response characteristics of IGZO-EGT. **a**, Schematic illustration of the neuromorphic transistor that can be stimulated using optical and electrical signals. The InGaZnO₃ film serves as a channel between the source (S) and drain (D) electrodes, and IL is used as the gating medium. **b**, Transfer curves of the IGZO transistor measured with $V_{SD} = 0.5$ V. **c**, The channel current controlled through a series of V_G pulse with a pulse width of 1 s and different amplitudes which shows the transition of the PSC from volatile to non-volatile property. **d**, Channel current under voltage stimulation of +1 V and +3 V. **e**, +1 V voltage pulse stimulation with different pulse durations. **f**, +2.5 V voltage pulse stimulation with different pulse durations. **g**, Pulse-switching characteristics. The high conductance state is generated by $V_G = +2.5$ V, and the low conductance state is generated by $V_G = -2.5$ V. **h**, The non-volatile multi-level conductance retention properties. The multi-states are produced by a series of V_G pulses with different pulse numbers ($V_G = +2.5$ V). **i**, Cyclic controlled LTP using V_G (equally spaced from +1.5 to +3.5 V, duration of 1 s, spaced 1 s apart) and LTD using V_G (equally spaced from -1.5 to -3.5 V, duration of 1 s, spaced 1 s apart) for 10 pulses.

Supplementary Figure 26. Photoresponse characteristics of IGZO-EGT. **a**, Evolution of channel current under UV light (375 nm) irradiation of different durations (7 mW/cm²). **b**, Evolution of channel current under blue light (450 nm) irradiation of different durations (70 mW/cm²). **c**, Evolution of channel current under green light (532 nm) irradiation of different durations (70 mW/cm²). **d-f**, The evolution of channel current stimulated by different numbers of light pulses, respectively under UV (375nm), blue (450nm) and green light (532nm) illumination.

We added “Moreover, the approach demonstrated in our study can be utilized to a broad range of materials, as long as its relaxation process under stimuli has nonlinear and volatile characteristics.” [Conclusions, page 15]

We added “The 15 nm a-IGZO film was grown on SiO₂/Si substrates at room temperature under O₂ pressure of 5 Pa through pulsed laser deposition. Afterwards, the film was annealed at 300°C for 1 hour. Pulsed laser deposition was equipped with a 308-nm XeCl excimer laser, with an energy density of about 1 J/cm² and a repetition of 3 Hz. The samples were cooled down to room temperature at 20 °C/min.” [Method, Sample preparation]

We added “XPS measurements were performed on ThermoFisher Scientific ESCALAB 250X under monochromatic Al K α radiation with an energy of 1486.6 eV.” [Method, Material characterization]

We further optimize the performance of BaSnO₃ devices and updated all of the figures with the newly prepared device.

Fig. 2| EGT with reconfigurable characteristics for multimode sensing. **a**, Schematic illustration of the neuromorphic transistor that can be stimulated using optical and electrical signals. The BaSnO₃ film serves as a channel between the source (S) and drain (D) electrodes, and IL is used as the gating medium. **b**, Short-term potential effect stimulated using optical pulses (1s) with a 0.5 V bias to monitor the channel current. **c**, Transition from STM to LTM under different light durations. **d**, Memory decay process after light pulse (70 mW/cm² for 1 s) stimulation, fitted using a double-exponential function. The fitting result shows the coexistence of two physical processes. **e**, Excitatory post synaptic current (PSC) induced via a train of optical pulse (70 mW/cm², 1s with intervals of 1, 5, 20 and 30s) with a 0.5 V bias to monitor the channel current. **f**, The channel current controlled through a series of V_G pulse with a pulse width of 1s and different amplitudes which shows the transition of the PSC from volatile to non-volatile property. **g**, Spike number-dependent plasticity under electrical stimuli (+2 V, 1s) with the same intervals of 1s. **h**, Cyclic controlled LTP using V_G (equally spaced from +1.5 to +3.5 V, duration of 1s, spaced 1s apart) and LTD using V_G (equally spaced from -1.5 to -3.5 V, duration of 1s, spaced 1s apart) for 32 pulses.

Fig. 3| Nonlinear mapping of multimodal signals based on EGT reservoirs. a, Combination mode for 16 inputs of 4-bit optical and electrical pulses. Here, both the optical and electrical pulse widths were 1 s and the pulse interval was 1 s. **b,** Response characteristics of the channel current in 1111, 1100, 1000, 1000 combinations under the “LLLE” mode input (light intensity 70 mW/cm², V_G = 1 V). **c,** The distinguishable output of 4-bit reservoir states reading at 0.3 V. **d,** Initial and final current values of the reservoir state with pulse stimulation of 16 combinations in the LLLE inputs mode.

Fig. 4 | Fused information input reservoir for recognition of multimodal Fashion-MNIST datasets. **a**, Image perception in complex environments. The right half of information can only be perceived through electrical signals, while the left half of information can only be perceived through visual signals. The original image ($28 \times 28 = 764$ pixels) was reorganized into a 196×4 by column vector array. **b**, Output results of the reservoir in the “LLLL” sensing mode, when 10% (top) and 90% (bottom) of the image can only be read by electrical signals. **c**, The change of ANN synaptic weight before and after training. The trained synaptic weight conforms to the normal distribution. **d**, **e**, Recognition accuracy under the five perception methods when 0% and 50% of the image can only be collected by electrical signals, **f**, Recognition accuracy of the five information perception modes variation with the proportion of the contaminated part (which can only be read via electrical signals).

Fig. 5| Mimics human perception of dynamic audiovisual information. a, Simplified diagram of humans achieving high levels of cognition through audiovisual integration. **b,** Three-dimensional space mapping of “Rolling hand toward left” gesture, a sample from the EgoGesture dataset. **c,** Images superimposed at four time points in the intermediate state after hand movements were preprocessed by the reservoir. **d,** Speech spectrogram of “Rolling hand toward left” gesture, reflecting the frequency domain distribution of audio at different moments. Corresponding voice overs for gestures, generated using text-to-speech engine. **e,** Input vector after normalization of time and frequency domain information, and output vector after reservoir processing. **f,** Comparison of recognition accuracy under single- and multi-mode information processing.

Supplementary Figure 5. The effect of UV light intensity on the channel current. Relaxation time of BaSnO₃ device under various light intensity with the same illumination durations 1s.

Supplementary Figure 6. Paired pulse facilitation (PPF) plasticity. **a**, Channel current is measured under UV irradiation with identical light intensity and duration time (light intensity of 70 mW/cm², the duration of 1 s). **b**, PPF ratio as a function of the pulse intervals, where the red line represents fitting results using the double exponential decay function.

Supplementary Figure 7. Influence of pulse time interval on channel current. **a**, Changes of channel current stimulated by UV pulses (light intensity of 70 mW/cm^2 , the duration of 1 s) with varying pulse intervals. **b**, Relationship between the channel current and the number of pulses, the current value after the pulse is applied for 1 s is used as the sampling point

Supplementary Figure 8. The effect of the light exposure on the channel current under blue laser irradiation. **a**, Device response to illumination of different wavelengths (70 mW/cm^2 , duration 1 s) under the same conditions. Among them, UV causes a more obvious change in conductance. **b**, I_{SD} response to blue light irradiation

at different durations (140 mW/cm^2). **c**, Spike number dependent plasticity under blue light irradiation with the same pulse width and intervals (140 mW/cm^2 for 1 s). **d**, Relationship between channel current and sequence pulse interval, the channel current is measured under light irradiation with different interval duration (light intensity of 140 mW/cm^2 , the duration of 1 s)

Supplementary Figure 11. Electrical performance of BaSnO₃ transistor. **a**, Transfer curve of the BaSnO₃ electrolyte-gated transistor (BSO-EGT) measured on $V_{SD}=0.5 \text{ V}$. The gate voltage was swept from 0 V to 2.5 V, 2.5 V to -2.5 V, and then back to 0 V. **b**, Output curve of BSO-EGT at a V_{SD} range of 0 to 1.5V.

Supplementary Figure 12. Long-term and short-term plasticity of BaSnO₃ transistors. **a**, Under the voltage stimulation of $V_G=1 \text{ V}$ (with different durations), the channel current can quickly return to the initial state. **b**, Under the electrical pulse stimulation of $V_G=2 \text{ V}$, the conductance of the device exhibits obvious non-volatile characteristics.

Supplementary Figure 13. Spike-frequency-dependent plasticity. Current change subject to 8 voltage pulse ($V_G=+1$ V, 1 s) with different pulse intervals.

Supplementary Figure 14. The non-volatile multi-level conductance retention properties. The multi-states are produced by a series of V_G pulses with different durations ($V_G=+3$ V, durations from 1 s to 5 s).

Supplementary Figure 15. Pulse-switching characteristics. The high conductance state is generated by $V_G = 2$ V, and the low conductance state is generated by $V_G = -2$ V. The pulse duration is 1 s. We choose 1 s after applying the voltage pulse stimulation as the sampling point.

Supplementary Figure 17. Device Characteristics for Mixed-Mode Input. **a**, Changes in channel current are monitored by a 0.5 V read voltage at the three mode inputs (“LL”, “LE”, “EE”). Among them, “LL” represents two optical pulses (70 mW/cm²), “LE” represents one optical pulse and one electric pulse, and “EE” represents two electric pulse signal inputs (1 V). The pulse width and time interval are both 1 s. **b**, Under “LE” mode input, the influence of the time interval between two pulses on the current value of the sampling point, the current value is collected 1 s after the second pulse is applied.

Supplementary Figure 18. Nonlinear mapping of 4-bit inputs based on the BSO reservoir. **a, b,** Changes in channel currents in response to light and electrical stimulation with different coding combinations, respectively. **c, d,** show the final distinguishability of the 16 encodings under light and electrical stimulation, respectively. The final distinguishable state is taken at 1 s after the application of the fourth pulse

Supplementary Figure 19. Multimodal nonlinear dynamics for reservoir computing. Variety of input waveform patterns. **a**, “LEEE” mode, **b**, “LLEE” mode demonstrate different current states distribution. **c**, The distinguishable output of 4-bit reservoir states of “LEEE” input. **d**, The distinguishable output of 4-bit reservoir states of “LLEE” input.

Corresponding references

58. Hays DC. *et al.* Energy band offsets of dielectrics on InGaZnO₄. *Applied Physics Reviews* 4, (2017).
59. Jang Y. *et al.* Amorphous InGaZnO (a-IGZO) Synaptic Transistor for Neuromorphic Computing. *Acs Applied Electronic Materials* 4, 1427-1448 (2022).
60. Ke S. *et al.* Indium-Gallium-Zinc-Oxide Based Photoelectric Neuromorphic Transistors for Modulable Photoexcited Corneal Nociceptor Emulation. *Advanced Electronic Materials* 7, 2100487 (2021).
61. Ide K. *et al.* Effects of excess oxygen on operation characteristics of amorphous In-Ga-Zn-O thin-film transistors. *Applied Physics Letters* 99, (2011).
62. Liu H. *et al.* High Performance and Hysteresis-Free a-IGZO Thin Film Transistors Based on Spin-Coated Hafnium Oxide Gate Dielectrics. *IEEE Electron Device Letters* 44, 1508-1511 (2023).

Point 2.

Furthermore, the incorporation of both electrical and optical signals, though methodologically sound, follows a well-tread path, with experiments echoing

conventional models like that of Pavlov's dog, thereby failing to exhibit marked progress or novel insights. [2]

Response:

Thanks for the reviewer's comments on our manuscript.

The fusion of optical and electrical information is not meaningless. We use it to represent visual and auditory information respectively, giving neuromorphic devices high-level cognitive capabilities. Perceptual enhancement of neural and behavioral response due to combinations of multisensory stimuli are found in many animal species^{R67, R68}. Behavioral and psychological experiments on mammals have shown that the integration of multisensory information can effectively improve perceptual performance, including neural and behavioral responses^{R69}. Current research on multi-modal integration often requires the use of multiple types of sensors and edge computing devices to implement the entire process^{R70, R71}. Benefiting from the multi-modal sensing, storage and processing performance of our proposed device, complete audio-visual integration and recognition tasks can be realized on a single device. The processing of dynamic tasks reflects the real-time processing capability of our proposed device .

We carefully read the reference given by the reviewer, which simulates a Pavlov's dog based on the device's ability to respond to light and electrical stimulation. But it is worth noting that phenomena more complex than Pavlov's dogs have been realized based on our devices. In essence, Pavlov's dog experiments defined a threshold that the current could not exceed when the device was stimulated by only electrical pulses, but could be exceeded by applying electrical stimulation after light stimulation. This is more like a binary problem, divided into whether it can exceed the threshold. In the application in Figure 4, we use a combination of light and electrical stimulation to encode the input signal so that the current of the device reaches different values, that is, 16 different states are generated, instead of just judging whether the threshold is reached. What's more important is that Pavlov's dog recommended by the reviewer does not use neural networks to identify and classify information. This is completely different from the message we convey in this manuscript. In Figure 5 we push the combination of light and electricity to deeper applications. Based on the nonlinear relaxation process exhibited by the device, a reservoir computing system was constructed to process dynamic information containing time correlation, which is difficult to achieve based on simple synaptic function devices. In this application, we combine dynamic video and audio information to imitate the process of human beings relying on vision and hearing to achieve high-level cognition of the external world. This will play an important role in future bionic and human-computer interaction platforms. Higher recognition accuracy than single-mode signals was verified. Further reservoir and subsequent readout networks are trained based on device characteristics.

Corresponding reference:

R67 Solvi C, *et al.* Bumble bees display cross-modal object recognition between visual and tactile senses. *Science* 367, 910-912 (2020).

R68. Jiang C, *et al.* Mammalian-brain-inspired neuromorphic motion-cognition nerve achieves cross-modal perceptual enhancement. *Nature Communications* 14, 1344 (2023).

R69. Zheng, Q., Zhou, L. & Gu, Y. Temporal synchrony effects of optic flow and vestibular inputs on multisensory heading perception. *Cell Reports*. 37, 109999 (2021).

R70. Wang M, *et al.* Gesture recognition using a bioinspired learning architecture that integrates visual data with somatosensory data from stretchable sensors. *Nature Electronics* 3, 563-570 (2020).

R71. Liu M, *et al.* A star-nose-like tactile-olfactory bionic sensing array for robust object recognition in non-visual environments. *Nature Communications* 13, 79 (2022).

Changes made:

We added “Benefiting from the multi-modal sensing, storage and processing performance of our proposed device, complete audio-visual integration and recognition tasks can be realized on a single device. The processing of dynamic tasks reflects the real-time processing capability of our proposed device.” [Conclusions, page 15]

Point 3.

The study does not explore the broader potential of neuromorphic devices beyond basic image transformation and data processing, notably omitting the exploration of high-resolution array configurations. [3] This limitation narrows the scope of the manuscript, diminishing its relevance and applicability in more advanced contexts that necessitate complex pattern recognition or extensive data processing capabilities.

Response:

Thank you very much for your efforts in reviewing the manuscript.

In recent years, researches on multi-modal recognition have received widespread attention^{R72, R73}. For example, the article published in *Nature Electronics* uses cameras and pressure sensors to achieve multi-modal perception of vision and gesture movements^{R74}. However, most current researches used different types of sensors to achieve multi-modal perception, and there is a lack of research on identifying different types of information using a single device. What’s more important is that information processing requires additional edge computing devices in some studies. These problems limit the miniaturization and integration of actual devices.

Our work goes beyond applying neuromorphic devices to basic image transformation and data processing tasks. In terms of potential applications of neuromorphic devices, we explored three aspects. First of all, the device’s response to a variety of external stimuli is used to implement multi-modal reservoir computing, which improves the ability to comprehensively obtain information in complex environments. More importantly, the reservoir’s unique advantages in processing dynamic information are

exploited to achieve recognition of speech and motion gestures. Last but not least, as a proof of concept, the audio-visual integration process is simulated on a single device, relying on the previously mentioned advantages. These all reflect the potential of our proposed device in the development process of future bionic platforms and human-machine interaction. However, it is difficult to achieve the above dynamic information recognition using simple synaptic devices.

We agree with the reviewer that actual device arrays will undoubtedly play an important role in practical applications. Following the literature recommended by the reviewer, we supplemented the device array experiments to demonstrate the practical application potential of BSO-EGT in more advanced contexts.

Figure R11. A 3×4 array for pattern learning. **a**, Three letters “I”, “O” and “P” are selected as inputs to the array. **b**, The device current (I_{SD}) was recorded after 15 UV pulse irradiations (70 mW/cm^2 , duration 2 s). **c**, Channel current distribution at 50 s after the end of the last light pulse. **(d)** Channel current distribution at 350 s after the end of the last light pulse.

Changes made:

We added “Image recognition and memory are important functions of artificial vision systems. To simulate a UV-sensitive vision system, we fabricated a 3×4 pixelated array using BSO-EGT. Three letters “I”, “O” and “P” are used to test the ability of the array to learn and remember images. Under the stimulation of 15 consecutive UV pulses, the image can still be clearly resolved after 350 s, which proves the advantages of our device in simulating the visual system. (Supplementary Figure 10)” [Paragraph 2, page 6]

Supplementary Figure 10. A 3×4 array for pattern learning. **a**, Three letters “I”, “O” and “P” are selected as inputs to the array. **b**, The device current (I_{SD}) was recorded after 15 UV pulse irradiations (70 mW/cm^2 , duration 2s). **c**, Channel current distribution at 50 s after the end of the last light pulse. **d**, Channel current distribution at 350 s after the end of the last light pulse.

Corresponding reference

R72. Wang M, *et al.* Gesture recognition using a bioinspired learning architecture that integrates visual data with somatosensory data from stretchable sensors. *Nature Electronics* 3, 563-570 (2020).

R73. Liu M, *et al.* A star-nose-like tactile-olfactory bionic sensing array for robust object recognition in non-visual environments. *Nature Communications* 13, 79 (2022).

R74. Zhu J, *et al.* A Heterogeneously Integrated Spiking Neuron Array for Multimode-Fused Perception and Object Classification. *Advanced Materials* 34, e2200481 (2022).

Point 4.

The authors shows different decay characteristics under light and electrical stimuli, crucial for nuanced state representations in data processing. However, the manuscript lacks a detailed explanation of these phenomena, impeding a thorough understanding of the device's intricate behavior. This gap is particularly concerning given the importance of these characteristics in practical applications.

Response:

Thanks for the reviewer's valuable on our manuscript. We are apologized for the lack of clarity in our description of the physical mechanisms. Based on the reviewer's questions, we would discuss the underlying mechanisms of the different behaviors of BSO-EGT under optical and electrical stimulation.

1. The volatile response of BSO-EGT under electrical stimulation originated from the electrical double layer (EDL). In this mechanism, ionic liquids are conductors of ions but are electrically classified as insulators. The rapid movement of anions and cations under an electric field produces a strong accumulation of space charge at the interface between the electrolyte and the channel, which exists in the form of an EDL, called a Helmholtz layer^{R75}. In theory established in 1853, an EDL can be viewed as a simple parallel plate capacitor. When a positive bias is applied to the gate, the anions in the ionic liquid move toward the interface between the electrolyte and the gate, due to the effect of the electric field^{R76}. At the same time, correspondingly, the cations in the ionic liquid migrate to the interface between the electrolyte and the channel to form a Helmholtz layer. Accumulation occurs due to blocking ions in the solid channel. In order to balance the EDL formed at the interface, an accumulation of electrons occurs inside the channel. Therefore, the conductance of the channel will change. When the external bias voltage is removed, the ions at the interface spontaneously migrate back into the ionic liquid, and the electrical double layer disappears, so the channel conductance returns to its original state.

2. The conductance of the BSO channel showed non-volatile changes under voltage stimulation, which was confirmed by experimental results (Figure R12a, b). The theoretical value of the electrolysis voltage of pure water is 1.23 V^{R77}. Considering the influence of experimental conditions, it will generally be higher than this value^{R78}. We found that this value is approximately 1.3 V in our experiments. Because at this voltage, there is a peak in the gate current of the transfer characteristic curve (Figure R12a), which is related to the hydrolysis reaction^{R79}.

Fig R12. a, Transfer curve of the BaSnO₃ electrolyte-gated transistor (BSO-EGT) measured on $V_{SD}=0.5$ V. **b**, Transfer curve for 15 consecutive scans between $V_G = -3V \sim +2.5V$. **c**, The channel current controlled through a series of V_G pulse with a pulse width of 1s and different amplitudes which shows the transition of the PSC from volatile to non-volatile property.

Below the threshold voltage, the device exhibits a volatile phenomenon due to the formation of EDL near the interface^{R80}, while under stimulation above the critical voltage, the channel conductance exhibits an obvious non-volatile phenomenon (Fig. R12c). This is because, in addition to the presence of EDL at the interface, H^+ originating from the trace water containing in the ionic liquids would be injected into the film when the positive voltage exceeded the critical voltage. Protons (H^+) produced by hydrolysis can be driven to the channel interior, resulting in strong interactions with the solid material^{R75, R76}. Therefore, the non-volatility of the device came from the migration of protons generated by hydrolysis in the ionic liquid into the oxide film under the positive gating, causing a non-volatile increase in the channel conductance.

To verify this mechanism, we performed secondary ion mass spectrometry (SIMS) experiments. The results (Fig. R13a) show that hydrogen ions appear inside the electrically modulated films, and the ion concentration rises significantly as the voltage increases. Moreover, the addition of a small volume of D₂O to the ionic liquids resulted in the presence of D⁺ signal in the electrically-modulated film, which was not observed inside the film without modulation (Fig. R13b). Therefore, this result clearly demonstrated that the inserted hydrogen ions under positive gating come from the trace water inside ionic liquids.

Fig R13. a, b, Secondary-ion mass spectrometry (SIMS) depth profiling in the pristine and gated BaSnO₃ films.

The non-volatile change in conductivity caused by ion migration (H^+ , Li^+) is not a unique conclusion, and has been reported in organic^{R81} and inorganic materials^{R82}. Since protons have a smaller ionic radius than other ions, they are more conducive to migrate into the interior of the film. The above mechanism has been demonstrated in many electrochemical transistors^{R79}.

2. The reversibility of the non-volatile behavior is also demonstrated in the revised manuscript. The transfer characteristic curve of the device was continuously measured 15 times, (Fig. R12b) and there was no obvious degradation in device performance. The conductance increase caused by ion migration can be restored to the initial state under the action of negative voltage, so the curves overlap well. When a positive pulse above the threshold voltage was applied to the device, a significant non-volatile increase in channel conductance occurs. Using negative gate voltage pulse stimulation, this process can proceed in the reverse direction. This was well verified during the LTP and LTD measurements (Fig. 2h), where we made the device reach a high conductance state by applying positive voltage stimulation and decreased the conductance by corresponding negative pulses. In the transfer characteristic curve, it can be seen that when a cycle scan ends, the conductance of the device returns to near the initial state. In brief, all the above experiments confirm that the device can be reset.

3. Previous studies have verified that oxygen vacancies will be generated in BSO films under ultraviolet (UV) irradiation, resulting in an increase of film conductance. Lee *et al.* used ambient-pressure X-ray photoemission spectroscopy (APXPS) for in-situ characterization (Figure R14) to monitor the chemical origin of generated defects induced by UV irradiation in real time^{R83}. O 1s spectra of BaSnO₃ epitaxial films were acquired, with UV-light illumination under vacuum and without illumination in oxygen atmosphere. After the UV light was illuminated on the BaSnO₃, the binding energy of O 1s shifted toward higher with development of an additional peak shoulder that

strengthened as illumination duration increased. Before the UV illumination under vacuum, the O 1s spectra could be mostly assigned to the peak related to lattice oxygen (O(I) at 529.9 eV) with a tiny fraction of the peak related to defective oxygen (O(II) at 531.3 eV)^{R18}. The peak shifts toward higher binding energy upon the illumination of UV light, the area fraction of O(II) peak gradually increased from 0.07 to 0.111 after 9600 s of illumination, indicating that the UV illumination under vacuum leads to the chemical modification by evolution of the oxygen-vacancy-related defects on the surface of BaSnO₃ (Fig. R14a). As a reverse process, *in situ* characterization using APXPS confirmed that exposure of oxygen-deficient BaSnO₃ films to oxygen atmosphere at 350 °C in darkness caused the area fraction of O(II) peak to decrease and gradually recover to the original states.

In addition, Lee *et al.* confirmed the existence of oxygen-related defects at the surface from cross-sectional scanning transmission electron microscopy (STEM) by comparing high-angle annular dark-field (HAADF) and low-angle annular dark-field (LAADF) signals in vacuum-illuminated BaSnO₃ epitaxial films with nearly perfect arrangement of atoms with cubic symmetry along the [100] zone axis (Figure R14b)^{R83}. The contrast-intensity profiles (yellow rectangles) show distinct change of contrast near the surface of vacuum-illuminated BaSnO₃ epitaxial films between HAADF and LAADF: contrary to almost identical HAADF contrast across the surface, LAADF contrast showed a higher contrast within ≈ 3 nm from the surface than in the bulk area. The difference occurs because the LAADF signal is more sensitive to the strain field from point defects (i.e., V_O) than the HAADF signal. This experiment also verified that the increase in the conductance of the BSO film under UV irradiation is related to the oxygen vacancies generated on the surface.

Our experimental phenomena are fully consistent with the results of Lee *et al.*' study. So it is reasonable to conclude that the conductance change produced by the BSO channel under UV light is caused by oxygen vacancies. When the UV dose [UV dose (mJ/cm²) = UV Intensity (mW/cm²) × Exposure Time (s)] is low, the oxygen vacancy concentration is low, and its impact on the BSO channel can be recovered quickly. On the other hand, when the UV dose is high, the oxygen vacancy concentration increases, resulting in non-volatile conductance changes in the BSO channel.

Fig R14. Synchrotron and STEM analysis of BaSnO₃ thin films during vacuum illumination^{R83}. **a**, In situ APXPS O 1s spectra of BaSnO₃ epitaxial films with ultraviolet-light illumination under vacuum at room temperature (left) and without illumination in oxygen atmosphere at 350 °C (right), with deconvoluted spectra related to lattice oxygen (O(I) at 529.9 eV) (gray shaded area) and defective oxygen (O(II) at 531.3 eV) (red shaded area). **b**, HAADF-STEM (left, top figure) and LAADF-STEM (left, bottom figure) image of vacuum-illuminated BaSnO₃ epitaxial films along the [100] zone axis (scale bar = 5 nm). Contrast-intensity profiles (right) from the yellow rectangles in HAADF-STEM (black line) and LAADF-STEM (red line) images.

Corresponding reference

- R75. Bisri SZ, *et al.* Endeavor of Iontronics: From Fundamentals to Applications of Ion-Controlled Electronics. *Advanced Materials* 29, 1607054 (2017).
- R76. Yuan H, *et al.* High-Density Carrier Accumulation in ZnO Field-Effect Transistors Gated by Electric Double Layers of Ionic Liquids. *Advanced Functional Materials* 19, 1046-1053 (2009).
- R77. Prasad B, *et al.* Integrated Circuits Comprising Patterned Functional Liquids. *Advanced Materials* 30, 1802598 (2018).
- R78. Ji H, Wei J, Natelson D. Modulation of the Electrical Properties of VO₂ Nanobeams Using an Ionic Liquid as a Gating Medium. *Nano Letters* 12, 2988-2992 (2012).
- R79. Lu N, *et al.* Electric-field control of tri-state phase transformation with a selective dual-ion switch. *Nature* 546, 124-128 (2017).

- R80. Nishioka D, *et al.* Edge-of-chaos learning achieved by ion-electron-coupled dynamics in an ion-gating reservoir. *Science Advances* 8, eade1156 (2022).
- R81. Wang S, *et al.* An organic electrochemical transistor for multi-modal sensing, memory and processing. *Nature Electronics* 6, 281-291 (2023).
- R82. Liang X, *et al.* Multimode transistors and neural networks based on ion-dynamic capacitance. *Nature Electronics* 5, 859-869 (2022).
- R83. Lee Y, *et al.* Reversible Manipulation of Photoconductivity Caused by Surface Oxygen Vacancies in Perovskite Stannates with Ultraviolet Light. *Advanced Materials* 34, e2107650 (2022).

Changes made:

To better elaborate the intrinsic mechanisms of BSO-EGT under different stimuli, we added “Previous studies have verified that oxygen vacancies will be generated in BSO films under UV irradiation, resulting in an increase of film conductance. Lee *et al.* used ambient-pressure X-ray photoemission spectroscopy (APXPS) for *in-situ* characterization to monitor the origin of generated defects and proved that the UV illumination under vacuum leads to chemical modification by evolution of the oxygen-vacancy-related defects on the surface of BaSnO₃³⁹. And the result of cross-sectional scanning transmission electron microscopy (STEM) in vacuum-illuminated BaSnO₃ epitaxial films also confirmed the existence of oxygen-related defects at the surface³⁹. When the UV dose [UV dose (mJ/cm²) = UV Intensity (mW/cm²) × Exposure Time (s)] is low, the oxygen vacancy concentration is low, and its impact on the BSO channel can be recovered quickly. On the other hand, when the UV dose is high, the oxygen vacancy concentration increases, resulting in non-volatile conductance changes in the BSO channel.” [Paragraph 3, page 6]

We changed “Supplementary Figure 15a shows that hydrogen ions appear inside the electrically modulated films, and the ion concentration rose significantly as the voltage increased. The addition of a small volume of D₂O to the IL resulted in the presence of D⁺ signal in the electrically-modulated film, which was not observed inside the film without modulation (Supplementary Figure 15b).” to “Supplementary Figure 15a shows that hydrogen ions appear inside the electrically modulated films, and the hydrogen ion concentration rose significantly as the voltage increased. We added a small volume of D₂O to the IL, in which D⁺ just acts as an isotope marker to show the source of the protons (H⁺). The addition of D₂O to the IL resulted in the presence of D⁺ signal in the electrically-modulated film, which was not observed inside the film without modulation (Supplementary Figure 15b).” [Paragraph 3, page 8]

We changed “For a positive V_G lower than the threshold value of the hydrolysis reaction, anions and cations in ILs accumulate at the IL/gate and IL/channel interfaces, respectively.³⁵ Thus, the introduction of electrons through EDL can increase channel conductance effectively.” to “The volatile response of BSO-EGT at a lower V_G originated from the rapid movement of anions and cations in ILs under an electric field, which produced a strong accumulation of space charge at the interface between the

electrolyte and the channel, called a Helmholtz layer or EDL⁴⁸. Accumulation occurs due to blocking ions in the solid channel. In order to balance the EDL formed at the interface, an accumulation of electrons occurs inside the channel. Therefore, the conductance of the channel will change. When the external bias voltage is removed, the ions at the interface spontaneously migrate back into the ionic liquid. The electrical double layer disappears, so the channel conductance returns to its original state.” [Paragraph 2, page 7]

We added “The conductance of the BSO channel showed non-volatile changes under high voltage stimulation, and there is a peak in the gate current of the transfer characteristic curve at approximately 1.3 V (Supplementary Figure 10a), which is related to the hydrolysis reaction⁴⁹. This is because, in addition to the presence of EDL at the interface, H⁺ originating from the trace water containing in the ionic liquids would be injected into the film when the positive voltage exceeded the critical voltage. Protons (H⁺) produced by hydrolysis can be driven to the channel interior, resulting in strong interactions with the solid material^{48,50}. Therefore, the non-volatility of the device came from the migration of protons generated by hydrolysis in the ionic liquid into the oxide film under the positive gating, causing a non-volatile increase in the channel conductance.” [Paragraph 2, page 8]

We changed “In order to analyze the origin of the non-volatility, secondary ion mass spectrometry (SIMS) was performed on the BSO films after applying different electrical stimuli.” to “To verify this mechanism, secondary ion mass spectrometry (SIMS) was performed on the BSO films after applying different electrical stimuli.” [Paragraph 3, page 8]

Corresponding references were added in the revised manuscript.

48. Bisri SZ, Shimizu S, Nakano M, Iwasa Y. Endeavor of Iontronics: From Fundamentals to Applications of Ion-Controlled Electronics. *Advanced Materials* 29, 1607054 (2017).

49. Lu N, *et al.* Electric-field control of tri-state phase transformation with a selective dual-ion switch. *Nature* 546, 124-128 (2017).

50. Yuan HT, Shimotani H, Tsukazaki A, Ohtomo A, Kawasaki M, Iwasa Y. Hydrogenation-Induced Surface Polarity Recognition and Proton Memory Behavior at Protic-Ionic-Liquid/Oxide Electric-Double-Layer Interfaces. *Journal of the American Chemical Society* 132, 6672-6678 (2010).

Overall, we have made great efforts to improve the manuscript by incorporating all the reviewers' comments and further highlighting the novelty of our work. We believe that the revised paper has met the required standard of *Nature Communications*, and are looking forward to a refreshed view of this reviewer.

REVIEWER COMMENTS

Reviewer #1 (Remarks to the Author):

The authors have largely addressed all the concerns. I do not have further comments.

Reviewer #2 (Remarks to the Author):

- Transistor characteristics in Fig 11 reveal a pretty poor on/off ratio (despite the improvement). The area of the transistors and corresponding gate leakage performance is unclear, in both cases the current should be stated in units of Amp/cm².
- Fig 2c is actuated with light, and has fewer conductance states than fig 2F actuated by voltage. However, the final results seem to give much poorer accuracy for EEEE than LLLL (Fig 4). Why ?
- Fig 3: Using both voltage and light input, all states are within 0.6nA to 1nA, the min/max conductance state are less 2 times. (See also Supplementary fig 18 with fairly similar margins). yet high accuracy. Why ?
- There seems to be inconsistency between figure 4d,e, and f. It is seen that 0% invisibility gives maximum accuracy of 40% for EEEE, yet for LEEE the figure is consistently above 90%. The authors claimed that multisensory inputs lead to higher accuracy, (bottom of page 5 in response document), however, there is no explanation for their abrupt results in Fig 4f for combinations or mixed signals, (even if EEE).
- The authors claim the non-volatility of the BSO-EGT comes from the injection and diffusion process of hydrogen ions originating from hydrolysis (at 2V). Surely that should be a cause for volatility rather than the opposite ? There seems no depth to this explanation.
- Fig R5c is not a proof of non-volatility. The x-axis is only 30 seconds (and reducing).
- On Page 2 of their response document the authors claim "Our as-prepared BSO-EGT is in a high-resistance state, the downward adjustment in the pristine state will cause significant leakage, making it difficult to perform RC based on the depression process".
- They claim they avoided leakage by operating in a high resistance state, but with a very low separation, yet they claim good accuracy, especially for light? Furthermore, surely, all states need to be represented, both high and low in equal measures (ie "0" and "1"), so they need to better explain their argument about leakage?
- On Pages 5, 7, 9: "sampling points per second" is not sufficient. They should state Volts/sec.
- Supplementary fig 11: Hysteresis direction absent.
- Overall their long-winded explanations and number of figures seems to have catapulted out of proportion (76 pages!), to make this attractive to a reader. Surely they can answer to the point ?

Reviewer #3 (Remarks to the Author):

The revisions have improved the manuscript and adequately addressed the reviewer's comment. Therefore, I agree to publish on the Nature Communications.

Response Letter to Reviewers

Reviewer #1 (Remarks to the Author):

The authors have largely addressed all the concerns. I do not have further comments.

Response:

We greatly appreciate the reviewer's recommendation for publication.

Reviewer #2 (Remarks to the Author):

Point 1

Transistor characteristics in Fig 11 reveal a pretty poor on/off ratio (despite the improvement). The area of the transistors and corresponding gate leakage performance is unclear, in both cases the current should be stated in units of Amp/cm².

Response:

Thank you for your valuable comments. We agree with the reviewer that the on/off ratio of our BSO reconfigurable transistor is not pretty enough compared with traditional transistor devices. However, we would like to point out that the present work focused on a reconfigurable brain-inspired transistor that integrates sensing, memory and processing functions. As shown in Table R1 (Table 1 in Supplementary Materials), the BSO transistor with an on/off ratio of 10^6 is at an advanced level compared with other reconfigurable multimodal transistors.

Ref.	On/off ratio	Optical reconfigurability	Electrical reconfigurability	Reservoir+Synapse	Multisensory integration	Task type
1	$> 10^3$	/	Yes	Yes	/	static
2	$> 10^6$	Potentially Yes	Yes	/	Tactile + Visual	static
3	$> 10^6$	/	Yes	Yes	/	Dynamic
4	$> 10^3$	/	/	/	Ocular+Vestibular	Dynamic
This work	$> 10^6$	Yes	Yes	Yes	Audio + Visual	Dynamic

Table R1. Comparison with state-of-the-art multimodal neuromorphic transistors.

The area of the channel is $50 \mu\text{m} \times 180 \mu\text{m}$ as mentioned in Methods section. When a +2.5 V gate voltage is applied, the channel current (I_{SD}) is $0.75 \mu\text{A}$ and the leakage current (I_G) is 0.8 nA (Supplementary Figure 11a). The difference between the two reaches three orders of magnitude, so the effect of leakage current can be neglected. The thickness of the channel is 10 nm . The current density of I_{SD} is $0.75 \mu\text{A} / (50 \mu\text{m} \times 10 \text{ nm}) = 150 \text{ Amp/cm}^2$. The BSO-EGT adopts a coplanar side gate structure, and we added the dimensions of the gate in Supplementary Figure 4. The gate area covered by

the ionic liquid is approximately $10\ \mu\text{m} \times 180\ \mu\text{m}$, so the current density of leakage can be expressed as^{R1} $0.8\ \text{nA} / (10\ \mu\text{m} \times 180\ \mu\text{m}) = 1.1 \times 10^{-5}\ \text{Amp}/\text{cm}^2$.

Corresponding reference

R1. Sarkar R, *et al.* Epi-Gd₂O₃-MOSHEMT: A Potential Solution Toward Leveraging the Application of AlGa_N/Ga_N/Si HEMT With Improved ION/IOFF Operating at 473 K. *IEEE Transactions on Electron Devices* 68, 2653-2660 (2021).

Changes made:

We added “When the gate bias is +2.5 V, the channel current is 0.75 μA and the leakage current is 0.8 nA. The difference between the two reaches three orders of magnitude, so the effect of leakage current can be neglected.” [Paragraph 2, Page 7]

We added the dimensions of the gate in Supplementary Figure 4.

Point 2

Fig 2c is actuated with light, and has fewer conductance states than fig 2F actuated by voltage. However, the final results seem to give much poorer accuracy for EEEE than LLLL (Fig 4). Why ?

Response:

Thank you for your valuable question. We chose Fashion-MNIST images partially contaminated by pigment as the application scenario^{R2}. The left side can only be sensed by light signals, whereas the right side necessitates the assistance of pressure sensors to convert tactile sensations into electrical signals (Figure 4a). Here, the contaminated part is defined as invisible information. The “pollution degree” or “invisible degree” indicates how many proportions of the pixels in the image are contaminated.

An invisibility of 0% means that the reservoirs of “LLLL” mode can acquire all the information from the image, while that of “EEEE” mode cannot obtain any content. Therefore, the “LLLL” mode is able to achieve a higher recognition accuracy as the classification accuracy is based on actual information present in the image. In contrast, the inference of the “EEEE” model is indistinguishable from random guessing, so its recognition is much lower (Fig. 4d). And the situation is completely opposite in the case

of 100% invisibility. When the two modes can obtain equal proportions of picture information (50% invisibility), their recognition accuracy become very close (Fig. 4e).

Additionally, it should be noted that Fig. 2c and Fig. 2f, respectively, illustrate the different conductance states of the BSO device under different optical/electric stimuli, demonstrating the transition from short-term memory (STM) to long-term memory (LTM). However, it should be noted that we utilized the conductance states demonstrated in Fig. 3c (“LLLE”), Supplementary Figure 18c, d (“LLLL”, “EEEE”), and Supplementary Figure 19c, d (“LEEE”, “LLEE”) for reservoirs. Reservoirs of all modes uses an equal quantity of conductance states.

Corresponding reference

R2. Liu, K., *et al.* An optoelectronic synapse based on α -In₂Se₃ with controllable temporal dynamics for multimode and multiscale reservoir computing. *Nat Electron* 5, 761–773 (2022).

Changes made:

We changed “ For pictures of Fashion-MNIST partially contaminated by pigments (Fig. 4a), the left half can be perceived through optical signals, while the right half needs the help of pressure sensors to convert the information that can be sensed through touch into electrical signals. ” to “ For pictures of Fashion-MNIST partially contaminated by pigments (Fig. 4a), the left side can be perceived through optical signals, whereas the right side necessitates the assistance of pressure sensors to convert tactile sensations into electrical signals. Here, the contaminated part is defined as invisible information.”
[Paragraph 2, Page 11]

We added “An invisibility of 0% means that the reservoirs of ‘LLLL’ mode can acquire all the information from the image, while that of ‘EEEE’ mode cannot obtain any content. Therefore, the ‘LLLL’ mode is able to achieve a higher recognition rate as the classification is based on the actual information, while the inference of the ‘EEEE’ model is indistinguishable from random guessing, so its recognition accuracy is much lower (Fig. 4d). And the situation is completely opposite in the case of 100% invisibility.” [Paragraph 1, Page 12]

Point 3

Fig 3: Using both voltage and light input, all states are within 0.6nA to 1nA, the min/max conductance state are less 2 times. (See also Supplementary fig 18 with fairly similar margins). yet high accuracy. Why ?

Response:

Thank you for your valuable question. Although the 16-state values of some modes are relatively close, their distributions are still distinguishable. Through computer simulation, many machine learning tasks can achieve high recognition accuracy^{R3- R5}. And we agreed with the reviewer that a larger distribution of conductance states will bring higher recognition accuracy to the neural network^{R6}. This is a future research direction for optimizing materials and designing device. In addition, this problem can be alleviated by amplifying the current in the hardware system through electronic devices such as trans-impedance amplifiers^{R7, R8}.

Corresponding reference

R3. Chen ZW, *et al.* All-ferroelectric implementation of reservoir computing. *Nature Communications* 14, 3585 (2023).

R4. Wu, X., *et al.* Wearable in-sensor reservoir computing using optoelectronic polymers with through-space charge-transport characteristics for multi-task learning. *Nature Communications* 14, 468 (2023).

R5. Sun L., *et al.* In-sensor reservoir computing for language learning via two-dimensional memristors. *Science Advance* 7, eabg1455 (2021).

R6. Wang, S., *et al.* An organic electrochemical transistor for multi-modal sensing, memory and processing. *Nature Electronics* 6, 281-291 (2023).

R7. Wang, W., *et al.* A memristive deep belief neural network based on silicon synapses. *Nature Electronics* 5, 870–880 (2022).

R8. Zhong, Y., *et al.* A memristor-based analogue reservoir computing system for real-time and power-efficient signal processing. *Nature Electronics* 5, 672–681 (2022).

Changes made:

We added “The distributions of 16-state values for different modes are distinguishable, so the classification tasks can ultimately be performed through computer simulation.”

[Paragraph 3, Page 10]

Point 4

There seems to be inconsistency between figure 4d,e, and f. It is seen that 0% invisibility gives maximum accuracy of 40% for EEEE, yet for LEEE the figure is consistently above 90%. The authors claimed that multisensory inputs lead to higher accuracy, (bottom of page 5 in response document), however, there is no explanation for their abrupt results in Fig 4f for combinations or mixed signals, (even if EEE).

Response:

Thank you for your valuable comments. As explained in our Response to Point 2, when the invisibility is 0%, “EEEE” mode cannot obtain any content. So, the inference of the “EEEE” model reservoir computing is indistinguishable from random guessing. Due to the consolidation of categories in the Fashion-MNIST database, the number of samples is not equal among the five clothing categories. The category “trousers” accounts for approximately 40% of the total samples. As a result, the EEEE mode achieves a recognition accuracy of around 40% after multiple iterations.

In contrast, the LLLL mode, LLLE mode, LLEE mode, and LEEE mode can retrieve 100%, 75%, 50%, and 25% of the information from the image, respectively, in a 0% visibility scenario. Therefore, these modes are able to get a higher recognition accuracy as the classification is based on the actual information of the images. Furthermore, the mixed mode, due to its utilization of two reading channels, can obtain partial information from the image under any visibility condition, thus achieving a higher recognition accuracy (Fig. 4f).

Changes made:

We changed “The single-signal modes do not facilitate the effective overall image information acquisition in particularly extreme cases, while the mixed-signal modes allow for judgment and classification based on global information.” to “ The single-signal modes do not facilitate the effective image information acquisition in particularly extreme cases, while the mixed-signal modes allow for judgment and classification based on actual information due to its utilization of two reading channels.” [Paragraph 2, Page 12]

Point 5

The authors claim the non-volatility of the BSO-EGT comes from the injection and diffusion process of hydrogen ions originating from hydrolysis (at 2V). Surely that should be a cause for volatility rather than the opposite? There seems no depth to this explanation.

Response:

Thank you for the valuable comments. In oxide material systems, there have been many studies on non-volatile changes in conductance due to ion insertion^{R9, R10}, among which the insertion of ions with small radii such as hydrogen ion is easier and more common^{R11, R12}. The non-volatile changes induced by proton injection can be realized through ionic liquid gating to hydrogenate oxide materials under electrical gating^{R13}. Subsequently, due to strong ion-lattice interactions, the hydrogenated phase can stably exist^{R14-R17}.

Besides, we conducted secondary ion mass spectrometry (SIMS) experiments. We observed an obvious proton concentration distribution inside the channel, indicating the existence of hydrogenation phenomenon. Additionally, there is a 12-hour waiting time for ionic liquid-gated samples before SIMS experiments are performed. This experiment also reveals that the insertion of hydrogen ions can exist inside the sample for a long time, leading to a non-volatile effect.

Corresponding reference

- R9. Lu Q, *et al.* Bi-directional tuning of thermal transport in SrCoOx with electrochemically induced phase transitions. *Nature Materials* 19, 655-662 (2020).
- R10. Tan AJ, *et al.* Magneto-ionic control of magnetism using a solid-state proton pump. *Nature Materials* 18, 35-41 (2019).
- R11. Shao, Z., *et al.* All-solid-state proton-based tandem structures for fast-switching electrochromic devices. *Nature Electronics* 5, 45–52 (2022).
- R12. Li, L., *et al.* Manipulating the insulator–metal transition through tip-induced hydrogenation. *Nature Materials*. 21, 1246–1251 (2022).
- R13. Leighton C. Electrolyte-based ionic control of functional oxides. *Nature Materials*. 18, 13-18 (2019).
- R14. Gao L, *et al.* Unveiling Strong Ion–Electron–Lattice Coupling and Electronic Antidoping in Hydrogenated Perovskite Nickelate. *Advanced Materials* 35, 2300617 (2023).

R15. Ji H, *et al.* Modulation of the Electrical Properties of VO₂ Nanobeams Using an Ionic Liquid as a Gating Medium. *Nano Letters* 12, 2988-2992 (2012).

R16. Bisri SZ, *et al.* Endeavor of Iontronics: From Fundamentals to Applications of Ion-Controlled Electronics. *Advanced Materials* 29, 1607054 (2017).

R17. Yuan H, *et al.* High-Density Carrier Accumulation in ZnO Field-Effect Transistors Gated by Electric Double Layers of Ionic Liquids. *Advanced Functional Materials* 19, 1046-1053 (2009).

Changes made:

We added “Additionally, there is a 12-hour waiting time for ionic liquid-gated samples before SIMS experiments are performed. Therefore, this experiment reveals that the insertion of hydrogen ions can exist inside the sample for a long time, leading to a non-volatile effect.” [Paragraph 2, Page 9]

Point 6

Fig R5c is not a proof of non-volatility. The x-axis is only 30 seconds (and reducing).

Response:

Thank you for the comments. Fig R5c is not for demonstrating the non-volatility of the BSO transistor. In fact, we used Fig R5c to show that the relaxation behavior of the channel conductance changes greatly as the amplitude of the applied voltage pulse increases. This phenomenon can simulate the transition from short-term memory (STM) to long-term memory (LTM) in biological systems.

We would like to stress that the non-volatile change in device conductance was demonstrated by Supplementary Figure 14 in our manuscript. With the stimulation of voltage pulses, the channel conductance undergoes multi-level non-volatile changes. Within 300 seconds, the conductance did not decline significantly. Non-volatile conductance changes at this time scale are sufficient to enable neuromorphic computing^{R18-R20}.

Corresponding reference

R18. Zhou F, *et al.* Optoelectronic resistive random access memory for neuromorphic vision sensors. *Nature Nanotechnology* 14, 776-782 (2019).

R19. Jiang T, *et al.* Tetrachromatic vision-inspired neuromorphic sensors with ultraweak ultraviolet detection. *Nature Communications* 14, 2281 (2023).

R20. Li, G. *et al.* Photo-induced non-volatile VO₂ phase transition for neuromorphic ultraviolet sensors. *Nature Communications*. 13, 1729 (2022).

Changes made:

We changed “There was no significant decay of channel conductance over a period of 300 s (Supplementary Figure 14).” to “There was no significant decay of channel conductance over a period of 300 s (Supplementary Figure 14), indicating that BSO-EGT has good non-volatility.” [Paragraph 2, Page 8]

Point 7

On Page 2 of their response document the authors claim “Our as-prepared BSO-EGT is in a high-resistance state, the downward adjustment in the pristine state will cause significant leakage, making it difficult to perform RC based on the depression process”. They claim they avoided leakage by operating in a high resistance state, but with a very low separation, yet they claim good accuracy, especially for light? Furthermore, surely, all states need to be represented, both high and low in equal measures (ie “0” and “1”), so they need to better explain their argument about leakage?

Response:

Thanks for the reviewer’s comments. In the previous response letter, the reviewer #1 concerned whether the depression process could also be used to perform reservoir computing (RC). In fact, as long as the relaxation process has nonlinear and volatile characteristics, in theory, both the potentiation (+V_G in our work) and depression (-V_G in our work) can be used to implement RC^{R21, R22}. In the negative gating (-V_G) region, the channel current is suppressed to be close to the leakage current and cannot be used for subsequent applications. Therefore, the potentiation process is more suitable than the depression process to perform RC in our BSO-EGT, which can avoid the influence of potential leakage current. This conclusion is well verified by the transfer characteristic curve (Supplementary Figure 11a). It should be noted that RC-related tasks in this work are carried out based on a positive gate voltage. Regarding the leakage current in the potentiation, as mentioned in our response to Point 1, it can be neglected compared to the channel current.

During the simulation process, encoding “0” and “1” reflect the input information (such as images, audio, etc.) after binary processing. In other words, they represent whether there is external stimulation (voltage pulses applied to the gate or light pulses irradiated on the channel) applied to the BSO-EGT. From this perspective, there is no relationship between binary encoding and leakage current. The output of the reservoir represents the channel current of the device after receiving a 4-bit encode input stream. At this point, as mentioned earlier, the effect of leakage current is negligible.

Corresponding reference

R21. Chen Z, *et al.* All-ferroelectric implementation of reservoir computing. *Nature Communications*. 14, 3585 (2023).

R22. Liu Z, *et al.* Interface-type tunable oxygen ion dynamics for physical reservoir computing. *Nature Communications*. 14, 7176 (2023).

Changes made:

Regarding the leakage issue, we have modified the manuscript accordingly (see Response to Point 1).

Point 8

On Pages 5, 7, 9: “sampling points per second” is not sufficient. They should state Volts/sec.

Response:

Thank you for your comments. The corresponding experiments on pages 5, 7, 9 were conducted under a constant reading voltage, so that “Volts/sec” cannot be used as the unit for scanning speed. Therefore, in the previous response letter, we adopted the “sampling points per second” to characterize the scanning speed in response to the reviewer’s question.

Point 9

Supplementary fig 11: Hysteresis direction absent.

Response:

We are grateful to the reviewer for this valuable comments. Following your suggestion, we have added arrows in the figure to indicate the direction of hysteresis (Fig. R1).

Fig. R1. Transfer curve of the BaSnO₃ electrolyte-gated transistor (BSO-EGT) measured on $V_{SD}=0.5$ V.

Changes made:

We added arrows in Supplementary Figure 11 to indicate the direction of hysteresis.

Point 10

Overall their long-winded explanations and number of figures seems to have catapulted out of proportion (76 pages!), to make this attractive to a reader. Surely they can answer to the point ?

Response:

We are very grateful to every reviewer for their time and efforts to improve our manuscript. Therefore, we always make every effort to respond to all comments and questions. Thank you very much!

Reviewer #3 (Remarks to the Author):

The revisions have improved the manuscript and adequately addressed the reviewer's comment. Therefore, I agree to publish on the Nature Communications.

Response:

We greatly appreciate the reviewer's recommendation for publication.

REVIEWERS' COMMENTS

Reviewer #1 (Remarks to the Author):

The authors have made efforts to address the new issues. I think that the response is overall ok. Hence, I would like to suggest the publication of this revised manuscript.

I notice that papers from the Nature family journals dominate the reference list. In fact, optoelectronic devices for neuromorphic computing have also been published in other journals, especially in early days.

Reviewer #3 (Remarks to the Author):

The revisions have improved the manuscript and adequately addressed the reviewer #2's comment. Therefore, I agree to publish on the Nature Communications.

Response Letter to Reviewers

Reviewer #1 (Remarks to the Author):

The authors have made efforts to address the new issues. I think that the response is overall ok. Hence, I would like to suggest the publication of this revised manuscript.

I notice that papers from the Nature family journals dominate the reference list. In fact, optoelectronic devices for neuromorphic computing have also been published in other journals, especially in early days.

Response:

We greatly appreciate the reviewer's recommendation for publication again. Following your suggestion, we have added relevant papers published in other journals.

Changes made:

We added related references [43] and [62]:

1. "However, as the duration gradually increased, the optical response of the device changed from volatility to non-volatility, similar to the short-term memory (STM) to long-term memory (LTM) transition of biological synapses⁴³"
2. "It occurs the band-to-band excitation, the oxygen vacancies ionization, and the metastable peroxides formation in the a-IGZO semiconductor during the light illumination^{61, 62}"

Corresponding references

43. Wang S, *et al.* A MoS₂/PTCDA Hybrid Heterojunction Synapse with Efficient Photoelectric Dual Modulation and Versatility. *Advanced Materials* **31**, 1806227 (2019).
62. Li HK, *et al.* A light-stimulated synaptic transistor with synaptic plasticity and memory functions based on InGaZnO_x-Al₂O₃ thin film structure. *Journal of Applied Physics* **119**, (2016).

Reviewer #3 (Remarks to the Author):

The revisions have improved the manuscript and adequately addressed the reviewer #2's comment. Therefore, I agree to publish on the Nature Communications.

Response:

We greatly appreciate the reviewer's recommendation for publication again.